# Deaths with COVID-19 and from all-causes following first-ever SARS-CoV-2 infection in individuals with preexisting mental disorders: A national cohort study from Czechia

Tomáš Formánek[1,2]*, Libor Potočár[1,3], Katrin Wolfova[4,5], Hana Melicharová[6], Karolína Mladá[1,7], Anna Wiedemann[2], Danni Chen[8], Pavel Mohr[9,10], Petr Winkler[1,11], Peter B. Jones[2‡], Jiří Jarkovský[6‡]

1 Department of Public Mental Health, National Institute of Mental Health, Klecany, Czechia, 2 Department of Psychiatry, University of Cambridge, Cambridge, United Kingdom, 3 PROMENTA Research Center, Department of Psychology, University of Oslo, Oslo, Norway, 4 Department of Epidemiology, Second Faculty of Medicine, Charles University, Prague, Czech Republic, 5 Department of Neurology, Columbia University Irving Medical Center, Columbia University, New York, New York, United States, 6 Institute of Health Information and Statistics of the Czech Republic, Prague, Czech Republic, 7 Department of Psychiatry, Faculty of Medicine in Pilsen, Charles University, Pilsen, Czech Republic, 8 Department of Clinical Epidemiology, Aarhus University, Aarhus, Denmark, 9 Clinical Center, National Institute of Mental Health, Klecany, Czech Republic, 10 Third Faculty of Medicine, Charles University, Prague, Czech Republic, 11 Health Service and Population Research Department, Institute of Psychiatry, Psychology and Neuroscience, King's College London, London, United Kingdom

‡ These authors are joint senior authors on this work.
* tf363@cam.ac.uk

**Data Availability Statement:** Due to legal regulations, individual-level patient data cannot be

## Abstract

### Background

Evidence suggests reduced survival rates following Severe Acute Respiratory Syndrome Coronavirus 2 (SARS-CoV-2) infection in people with preexisting mental disorders, especially psychotic disorders, before the broad introduction of vaccines. It remains unknown whether this elevated mortality risk persisted at later phases of the pandemic and when accounting for the confounding effect of vaccination uptake and clinically recorded physical comorbidities.

### Methods and findings

We used data from Czech national health registers to identify first-ever serologically confirmed SARS-CoV-2 infections in 5 epochs related to different phases of the pandemic: 1st March 2020 to 30th September 2020, 1st October 2020 to 26th December 2020, 27th December 2020 to 31st March 2021, 1st April 2021 to 31st October 2021, and 1st November 2021 to 29th February 2022. In these people, we ascertained cases of mental disorders using 2 approaches: (1) per the International Classification of Diseases 10th Revision (ICD-10) diagnostic codes for substance use, psychotic, affective, and anxiety disorders; and (2) per ICD-10 diagnostic codes for the above mental disorders coupled with a prescription for anxiolytics/hypnotics/sedatives, antidepressants, antipsychotics, or stimulants per the

publicly deposited. For sharing the data, access must be granted by IHIS, the curator of Czech national health registers. Access to data can be requested through the procedure described at the web page of IHIS: https://www.uzis.cz/index-en.php?pg=contact–provision-of-information. The full analytical code is available at a dedicated GitHub repository: https://github.com/libpot/COVID_Mortality_Preexisting_Mental_Disorders. HM and JJ had full access to all registers and take responsibility for the integrity of the data export. TF, LP, HM, KM, and JJ had full access to all exported data. TF and LP responsibility for the accuracy of the data analysis.

**Funding:** This study was supported by the Czech Health Research Council (grant number NU22-D-146; TF, HM, KM, PW, and JJ), the Ministry of Health, Czech Republic (funding number 00023752; TF, LP, KM, PM, and PW), and the National Institute for Health and Care Research (NIHR) Applied Research Collaboration East of England at Cambridgeshire and Peterborough NHS Foundation Trust (TF). The funders had no role in study design, data collection and analysis, decision to publish, or preparation of the manuscript. The views expressed are those of the authors and not necessarily of the NIHR, the Department of Health and Social Care or other funders.

**Competing interests:** The authors have declared that no competing interests exist.

**Abbreviations:** aHR, adjusted hazard ratio; ATC, Anatomical Therapeutic Chemical; CCI, Charlson Comorbidity Index; CI, confidence interval; COVID-19, Coronavirus Disease 2019; HR, hazard ratio; ICD-10, International Classification of Diseases 10th Revision; IQR, interquartile range; ISID, Information System of Infectious Diseases; NHIS, National Health Information System; NRRHS, National Registry of Reimbursed Health Services; SARS-CoV-2, Severe Acute Respiratory Syndrome Coronavirus 2; SD, standard deviation.

Anatomical Therapeutic Chemical (ATC) classification codes. We matched individuals with preexisting mental disorders with counterparts who had no recorded mental disorders on age, sex, month and year of infection, vaccination status, and the Charlson Comorbidity Index (CCI). We assessed deaths with Coronavirus Disease 2019 (COVID-19) and from all-causes in the time period of 28 and 60 days following the infection using stratified Cox proportional hazards models, adjusting for matching variables and additional confounders. The number of individuals in matched-cohorts ranged from 1,328 in epoch 1 to 854,079 in epoch 5. The proportion of females ranged from 34.98% in people diagnosed with substance use disorders in epoch 3 to 71.16% in individuals diagnosed and treated with anxiety disorders in epoch 5. The mean age ranged from 40.97 years (standard deviation [SD] = 15.69 years) in individuals diagnosed with substance use disorders in epoch 5 to 56.04 years (SD = 18.37 years) in people diagnosed with psychotic disorders in epoch 2. People diagnosed with or diagnosed and treated for psychotic disorders had a consistently elevated risk of dying with COVID-19 in epochs 2, 3, 4, and 5, with adjusted hazard ratios (aHRs) ranging from 1.46 [95% confidence intervals (CIs), 1.18, 1.79] to 1.93 [95% CIs, 1.12, 3.32]. This patient group demonstrated also a consistently elevated risk of all-cause mortality in epochs 2, 3, 4, and 5 (aHR from 1.43 [95% CIs, 1.23, 1.66] to 1.99 [95% CIs, 1.25, 3.16]). The models could not be reliably fit for psychotic disorders in epoch 1. People diagnosed with substance use disorders had an increased risk of all-cause mortality 28 days postinfection in epoch 3, 4, and 5 (aHR from 1.30 [95% CIs, 1.14, 1.47] to 1.59 [95% CIs, 1.19, 2.12]) and 60 days postinfection in epoch 2, 3, 4, and 5 (aHR from 1.22 [95% CIs, 1.08, 1.38] to 1.52 [95% CIs, 1.16, 1.98]). Cases ascertained based on diagnosis of substance use disorders and treatment had increased risk of all-cause mortality in epoch 2, 3, 4, and 5 (aHR from 1.22 [95% CIs, 1.03, 1.43] to 1.91 [95% CIs, 1.25, 2.91]). The models could not be reliably fit for substance use disorders in epoch 1. In contrast to these, people diagnosed with anxiety disorders had a decreased risk of death with COVID-19 in epoch 2, 3, and 5 (aHR from 0.78 [95% CIs, 0.69, 0.88] to 0.89 [95% CIs, 0.81, 0.98]) and all-cause mortality in epoch 2, 3, 4, and 5 (aHR from 0.83 [95% CIs, 0.77, 0.90] to 0.88 [95% CIs, 0.83, 0.93]). People diagnosed and treated for affective disorders had a decreased risk of both death with COVID-19 and from all-causes in epoch 3 (aHR from 0.87 [95% CIs, 0.79, 0.96] to 0.90 [95% CIs, 0.83, 0.99]), but demonstrated broadly null effects in other epochs. Given the unavailability of data on a number of potentially influential confounders, particularly body mass index, tobacco smoking status, and socioeconomic status, part of the detected associations might be due to residual confounding.

## Conclusions

People with preexisting psychotic, and, less robustly, substance use disorders demonstrated a persistently elevated risk of death following SARS-CoV-2 infection throughout the pandemic. While it cannot be ruled out that part of the detected associations is due to residual confounding, this excess mortality cannot be fully explained by lower vaccination uptake and more clinically recorded physical comorbidities in these patient groups.

## Author summary

### Why was this study done?

- Existing research has demonstrated consistently elevated risk of death with Coronavirus Disease 2019 (COVID-19) or all-cause mortality in people with preexisting psychotic and substance use disorders following a Severe Acute Respiratory Syndrome Coronavirus 2 (SARS-CoV-2) infection.

- The evidence on people with preexisting affective and anxiety disorders is broadly consistent with increased mortality risk; however, multiple studies demonstrated null effects.

- To the best of our knowledge, no study has used national data covering almost all inpatient and outpatient settings, including primary care, and laboratory-confirmed SARS-CoV-2 infections to investigate whether this elevated mortality risk was present throughout the pandemic, including its later phases, and when robustly accounting for the confounding effect of vaccination uptake and clinically recorded physical comorbidities.

### What did the researchers do and find?

- Using Czech national, whole population, all healthcare encompassing register-based data, individuals with preexisting psychotic and, less consistently, substance use disorders had increased risk of death with COVID-19 and all-cause mortality, including at the later phases of the pandemic.

- People with preexisting anxiety disorders had decreased risk of death with COVID-19 and all-cause mortality in multiple epochs, whereas people with preexisting affective disorders demonstrated broadly null effects throughout the pandemic.

- These associations could not be fully explained by differences in vaccination uptake or clinically recorded physical comorbidities.

### What do these findings mean?

- The consistently lower survival in people with preexisting psychotic and substance use disorders aligns with existing evidence on fatal health inequalities in these patient groups.

- Systemic efforts are needed to fully reverse the risk attributable to long-term, structural processes affecting health of people with psychotic and substance use disorders.

- The main limitation of the present study was its inability to fully control for a number of characteristics, particularly body mass index, tobacco smoking status, and socioeconomic status, that might confound the associations between mental disorders and mortality: future studies should explore these associations while accounting for these confounders.

## Introduction

Evidence before the outbreak of the Coronavirus Disease 2019 (COVID-19) pandemic showed that people with mental disorders have a higher risk of developing a wide range of physical health conditions relative to their counterparts without these disorders [1–3] as well as higher mortality rates and shorter life expectancies than the general population [2,4–7]. Worse general health, often associated with lower socioeconomic status and lifestyle risk factors (e.g., smoking) could contribute to an increased risk of Severe Acute Respiratory Syndrome Coronavirus 2 (SARS-CoV-2) infection and potentially lower survival following the infection in these people.

Previous research has demonstrated that individuals with a diagnosis of a mental disorder had an increased risk for SARS-CoV-2 infection [8] as well as for breakthrough infection after vaccination [9]. Existing evidence on mortality postinfection, then, showed consistently increased risk in people with psychotic [10–17] and substance use disorders [18–21], elevated risk [11,13,16,22] or null effects [15,18] in people with anxiety disorders, and elevated risk [11,14,16,23] or null effects [15] in people with affective disorders.

While this evidence suggests lower survival following SARS-CoV-2 infection in people with preexisting mental disorders, to the best of our knowledge, no study has used national data covering almost all inpatient and outpatient settings, including primary care, and laboratory-confirmed SARS-CoV-2 infections to investigate whether this elevated mortality risk persisted at later phases of the pandemic and when robustly accounting for the confounding effect of vaccination uptake and clinically recorded physical comorbidities.

In the present study, we used national, whole population, all healthcare encompassing register-based data to investigate the risk of death with COVID-19 and from all-causes following first-ever laboratory-confirmed infection with SARS-CoV-2 in individuals with preexisting mental disorders compared with matched counterparts without mental disorders at 5 distinct pandemic phases. By performing matching on vaccination status and clinically recorded physical comorbidities, we aimed to explore associations not confounded by differences between people with and without preexisting mental disorders on these characteristics.

## Methods

The research questions and the analytical plan were preregistered at Open Science Framework before data analyses started [24]: any deviations from the plan are described in S1 Methods. This study was reported as per the Reporting of studies Conducted using Observational Routinely collected health Data (RECORD) Statement (see S1 RECORD Checklist).

### Setting

Mental health care in Central and Eastern European region relies on large psychiatric hospitals [25–27]. Considering Czechia in particular, more than 50% of its mental health budget is allocated to inpatient services [28], with the majority of inpatient care provided in outdated psychiatric hospitals [29]. However, Czechia has launched its mental health reform in 2013, with its initial main goals focusing on deinstitutionalization. This entails the expansion of community-based services, alongside a reduction in long-term inpatient beds and complemented by educational, destigmatization and other implementation programs aimed at improving the quality of care and overall quality of life of people with psychiatric conditions [27,30].

The first wave of the pandemic in Czechia lasted roughly from 1st March 2020 to 30th September 2020, with a State of Emergency being in place from 12th March 2020 to 17th May 2020. The first wave resulted in 70,968 incident infections [31]. The second wave of the

pandemic lasted approximately from 1st October 2020 to 31th March 2020, with a State of Emergency imposed from 5th October 2020 to 11th April 2021. The second wave led to 1,482,727 incident infections [31]. The Czech National Vaccination Strategy was launched in December 2020, with preexisting mental disorders not considered as reason for priority inoculation [32]. The period of post-second wave lasted roughly from 1st April 2021 to 31st October 2021, resulting in 227,827 incident infections [31]. Then, the 2021 to 2022 winter wave lasted from approximately 1st November 2021 to 28th February 2022, with a State of Emergency imposed from 26th November to 25th December, and led to 1,820,446 incident infections [31].

## Ethics statement

This study was approved by the Ethics Committee of the National Institute of Mental Health (approval number 176/21).

## Data

We used data from the National Registry of Reimbursed Health Services (NRRHS), part of the National Health Information System (NHIS), covering inpatient and outpatient services, including primary care, as well as prescription medications. The register covers nearly the entire Czech population (approximately 10.7 million inhabitants). The records are created by health professionals who complete information on diagnosis (primary and secondary diagnoses) as per the International Classification of Diseases 10th Revision (ICD-10), date (for inpatient settings, admission, and discharge date), Anatomical Therapeutic Chemical (ATC) classification codes for prescription medications (with the exception of common medications administered in inpatient settings), and basic sociodemographic information such as age, sex, and region of permanent residency. Additionally, we used data from the Information System of Infectious Diseases (ISID) covering nationwide testing for SARS-CoV-2 and COVID-19 vaccination status. Furthermore, we used data from the register of all-cause mortality, containing information on the date of death, the ICD-10 cause(s), and, if applicable, the external cause(s) of death. All 3 registers can be interlinked using a common unique identifier and are maintained by the state-funded Institute of Health Information and Statistics of Czechia (IHIS). Data in the NHIS are collected in accordance with Act No. 372/2011 Coll., on health services and conditions of their provision, while ISID data are collected in accordance with Act No. 258/2000 Coll., on public health protection. Due to this legal mandate, the retrospective analyses of data in these registries did not require informed consents from participants.

We retrieved all individuals aged 10 or above—the earliest plausible onset age of the studied mental disorders [33]—with first-ever laboratory-confirmed SARS-CoV-2 infection occurring in 5 epochs:

1. 1st March 2020 to 30th September 2020, the first wave of the pandemic.

2. 1st October 2020 to 26th December 2020, the second wave of the pandemic before the initiation of the national vaccination program.

3. 27th December 2020 to 31st March 2021, the beginning of the national vaccination program to the end of the second wave of the pandemic.

4. 1st April 2021 to 31st October 2021, the post-second wave period.

5. 1st November 2021 to 28th February 2022, the 2021 to 2022 winter wave.

## Exposure

In individuals with first-ever laboratory-confirmed SARS-CoV-2 infection, we used 2 approaches to ascertain cases. The first approach relied on identifying the occurrence of diagnosis per ICD-10 codes for (1) substance use disorders (F1); (2) psychotic disorders (F2); (3) affective disorders (F3); and (4) anxiety disorders (F4) in the period of 5 years prior to the date of infection (see details in S1 Methods). We considered the occurrence of at least one of the above codes as any mental disorder. We established the occurrence of each of the mental disorders separately. We considered an individual to have a diagnosis when the given ICD-10 code was listed on a record in either inpatient (primary diagnosis, considered from discharge date) or any outpatient setting. Conversely, the unexposed cohort included individuals who had no such occurrence in the period of 5 years before the date of their SARS-CoV-2 infection.

The second approach entailed establishing whether an individual was prescribed psychopharmaceuticals at least once in the period of 5 years prior to the date of SARS-CoV-2 infection, in addition to occurrence of diagnosis (inpatient or any outpatient setting) for a given ICD-10 code. We considered the prescription per ATC codes of any anxiolytics/hypnotics/sedatives (N05B, N05C), antidepressants (N06A), antipsychotics (N05A), or stimulants (N06B). Conversely, the unexposed cohort included individuals who had no diagnosis of a mental disorder and no prescription of any psychopharmaceutical in the period of 5 years before the date of their infection.

We used the 2 ascertainment approaches to investigate the consistency of estimates across different exposure definitions: broadly consistent results between these would increase the confidence in the robustness of inferences.

## Control of confounding

In our identification and selection of potential confounders, we followed the "disjunctive cause criterion," in which one controls for covariates that are causes of the exposure or causes of the outcome or causes of both [34,35].

## Matching

In the first 2 epochs, we matched on age, sex, month and year of infection as well as the Charlson Comorbidity Index (CCI) [36]. In the 3 subsequent epochs, we matched on age, sex, month and year of infection, vaccination status, and the CCI. Since vaccination does not confer an immediate protection, we did not consider vaccinations that were administered 14 or less days before the infection. For example, when an individual received the first dose of a two-dose regimen more than 14 days before the infection, and the second dose 14 or less days before the infection, we considered them as having received the first dose at the time of the infection. The CCI referred to the period of 5 years before the date of the SARS-CoV-2 infection and was coded as 0, 1, 2, 3, and 4 or more comorbidities. Each exposed individual was matched with up to 5 unique unexposed counterparts. Some people with preexisting mental disorders had no matching counterparts (in each cohort <10%, see details in S1–S5 Tables); we excluded these unmatched individuals from the respective analyses.

## Additional confounders

To further reduce the level of unaccounted for confounding, we adjusted for region of permanent residency, overall number of contacts with inpatient services and overall number of contacts with outpatient services (disregarding contacts related to the exposure), and prescription medications (see the detailed list with ATC codes in S1 Methods). We considered prescription

of each of the medications or treatment administration in the period of 1 year prior to the SARS-CoV-2 infection separately. The number of contacts with the healthcare system referred to the period of 5 years before the date of the SARS-CoV-2 infection. For details, see the proposed directed acyclic graph in S1 Fig in S1 File.

## Outcome

We considered (1) deaths with COVID-19 (ICD-10 codes U071 and U072 listed as a cause of death on the death certificate); and (2) all-cause mortality occurring in the period of (1) 28 days; and (2) 60 days after a positive test for SARS-CoV-2. These cutoffs are based on Public Health England's analysis that showed that 88% and 96% of deaths occurred within 28 and 60 days of a positive test, respectively [37].

## Statistical analysis

Following descriptive analysis, we used stratified Cox proportional hazards models to assess the risks of deaths with COVID-19 and from all-causes in individuals with preexisting mental disorders compared with matched counterparts without such disorders, separately for each studied mental disorder and epoch. Each stratum consisted of 1 person with preexisting mental disorder and up to 5 matched counterparts. Time-to-event was expressed in days. In models investigating the risk of death with COVID-19, we considered death due to any other cause as competing risk, and the affected individuals were censored. We fitted models adjusting for confounders, with the CCI used as a continuous measure. The results were expressed as hazard ratios (HRs) with 95% confidence intervals (95% CIs). We tested the proportionality assumption using Schoenfeld residuals; in some instances, the assumption was violated, we therefore interpreted the HRs as weighted averages of the time-varying HRs over the entire follow-up period [38]. In line with the statement from the American Statistical Association on $p$-values [39], we present effect sizes with 95% CIs throughout the manuscript. However, we provide $p$-values as complementary information in Supplementary Results. All analyses were conducted in R statistical programming language (version 4.2.2) [40], using the libraries *survival* (version 3.5–5) and *EValue* (version 4.1.3) [41].

## Sensitivity analyses

Having a history of a mental disorder might influence the risk of being tested for SARS-CoV-2 infection; thus, restricting the analysis to individuals who had a positive test might lead to collider bias [42]. To examine potential presence of collider bias, we conducted negative control exposure analyses by assessing the associations between the characteristics that are expected to be unrelated to the outcome and the outcome itself within the chosen cohorts [42]. To do so, we considered the occurrence of (1) migraine (ICD-10 code G43); (2) fracture of forearm (ICD-10 code S52); (3) acne (ICD-10 code L70); (4) mild allergies (ICD-10 codes J301, L500, and L23); and (5) transport accidents (ICD-10 codes V01-V99) in the time period of 5 years prior to the positive SARS-CoV-2 test. We assessed the occurrence of each of these separately. Then, we fitted stratified Cox proportional hazards models with the above characteristics being the exposures, while using the confounders and outcomes from the main analysis. We reported the total number and proportion of non-null tests, with the theoretical maximum being 40 (or 100%) per 1 epoch-mental disorder combination (i.e., 2 case ascertainment definitions X 5 negative control exposures X 4 outcomes). Since some of our cohorts were considerably large (i.e., anxiety disorders in epochs 2 to 5), it would be possible to have non-null results even if the effect sizes were negligible [43]. Thus, we complementarily provided averaged HRs

across the 40 tests per 1 epoch-mental disorder combination. Proportion of non-null tests closer to 0% and/or averaged HRs closer to 1 would suggest the absence of collider bias.

To assess the level of potential unmeasured confounding, we calculated *E*-values for each of our regression model where the results were not consistent with a null effect. The *E*-value indicate the strength of association—here expressed in hazard ratio—an unmeasured confounder, or set of confounders, would need to have with both the exposure and the outcome to nullify the association between the exposure and the outcome observed in the model [44]. *E*-values closer to 1 indicate lower confidence in the results not being due to residual confounding [44].

## Results

The number of individuals in matched-cohorts ranged from 1,328 in epoch 1 (247 diagnosed and treated with psychotic disorders and 1,081 counterparts) to 854,079 in epoch 5 (150,211 diagnosed with anxiety disorders and 703,868 counterparts). The proportion of females ranged from 34.98% in people diagnosed with substance use disorders in epoch 3 to 71.16% in individuals diagnosed and treated with anxiety disorders in epoch 5. The mean age ranged from 40.97 years (standard deviation [SD] = 15.69 years) in individuals diagnosed with substance use disorders in epoch 5 to 56.04 years (SD = 18.37 years) in people diagnosed with psychotic disorders in epoch 2. The detailed descriptive statistics are provided in Tables 1 and 2 and those with additional confounders in S6 and S7 Tables.

### Risk of death with COVID-19 in people with preexisting mental disorders

In the models adjusting for all considered confounders, including vaccination uptake and clinically recorded physical comorbidities, people diagnosed with or diagnosed and treated for psychotic disorders had an elevated risk of death with COVID-19 in epochs 2, 3, 4, and 5, both 28 and 60 following SARS-CoV-2 infection. The models could not be reliably fit for psychotic disorders in epoch 1. Those diagnosed with substance use disorders had an increased risk of death with COVID-19 28 days postinfection in epoch 3 and 4 and 60 days postinfection in epoch 3. Cases ascertained based on diagnosis of substance use disorders and treatment by psychopharmaceuticals had an elevated risk of death with COVID-19 in epoch 3, both 28 and 60 days following infection. The models could not be reliably fit for substance use disorders in epoch 1, and the remaining ones were consistent with a null effect.

In contrast, people diagnosed with or diagnosed and treated for anxiety disorders had a decreased risk of death with COVID-19 in epoch 2, 3, and 5, both 28 and 60 days postinfection. The remaining models for anxiety disorders were consistent with a null effect. Additionally, people diagnosed and treated for affective disorders had a decreased risk of death with COVID-19 in epoch 3, both 28 and 60 days postinfection, but all other models involving affective disorders were broadly consistent with a null effect. The results for any studied mental disorder were—regardless of case ascertainment definition—consistent with a null effect in all epochs. For detailed results, see Figs 1 and S2–101 in S1 File and S8–S10 Tables.

### Risk of all-cause mortality in people with preexisting mental disorders

In the models adjusting for all considered confounders, including vaccination uptake and clinically recorded physical comorbidities, people diagnosed with or diagnosed and treated for psychotic disorders were more likely to die in epochs 2, 3, 4, and 5, both 28 and 60 days postinfection. The models could not be reliably fit for psychotic disorders in epoch 1. In those diagnosed with substance use disorders, there was an elevated risk of all-cause mortality 28 days postinfection in epoch 3, 4, and 5 and 60 days postinfection in epoch 2, 3, 4, and 5. Cases ascertained based on diagnosis of substance use disorders and treatment had increased risk of all-

cause mortality in epoch 2, 3, 4, and 5, both 28 and 60 days postinfection. The models could not be reliably fit for substance use disorders in epoch 1, and the remaining ones were consistent with a null effect.

Conversely, people diagnosed with anxiety disorders had a decreased risk of all-cause mortality in epoch 2, 3, 4, and 5, both 28 and 60 days postinfection. Cases ascertained based on diagnosis of anxiety disorders and treatment by psychopharmaceuticals demonstrated broadly

**Table 1. Descriptive statistics per matching variables, cases ascertained by diagnosis per the ICD-10 diagnostic codes.**

| Epoch | Characteristic | Any mental disorder | | Substance use disorders | | Psychotic disorders | | Affective disorders | | Anxiety disorders | |
|---|---|---|---|---|---|---|---|---|---|---|---|
| | | unexposed | exposed | unexposed | exposed | unexposed | exposed | unexposed | exposed | unexposed | exposed |
| 1 | Total, *n* | 29,549 | 7,274 | 3,670 | 789 | 1,275 | 271 | 7,601 | 1,647 | 24,925 | 5,797 |
| | Age, mean (SD) | 42.45 (17.38) | 44.36 (17.95) | 41.12 (17.62) | 42.25 (18.27) | 49.44 (19.81) | 50.39 (20.20) | 47.32 (17.06) | 48.17 (17.42) | 42.12 (17.10) | 43.43 (17.54) |
| | Sex, *n* (%) | | | | | | | | | | |
| | Females | 18,028 (61.01) | 4,684 (64.39) | 1,476 (40.22) | 317 (40.18) | 738 (57.88) | 156 (57.56) | 5,026 (66.12) | 1,092 (66.30) | 16,134 (64.73) | 3,892 (67.14) |
| | Infection month, median (IQR) | 9 (1) | 9 (1) | 9 (2) | 9 (2) | 9 (1) | 9 (1) | 9 (1) | 9 (1) | 9 (1) | 9 (1) |
| | Infection year, median (IQR) | 2020 (0) | 2020 (0) | 2020 (0) | 2020 (0) | 2020 (0) | 2020 (0) | 2020 (0) | 2020 (0) | 2020 (0) | 2020 (0) |
| | CCI, mean (SD) | 0.92 (1.55) | 1.2 (1.78) | 1.17 (1.8) | 1.33 (1.98) | 1.44 (2.04) | 1.58 (2.11) | 1.32 (1.82) | 1.47 (1.93) | 0.93 (1.53) | 1.14 (1.71) |
| 2 | Total, *n* | 322,884 | 72,815 | 40,742 | 8,160 | 21,480 | 4,300 | 86,932 | 17,396 | 26,6089 | 5,5758 |
| | Age, mean (SD) | 48.73 (18.28) | 49.48 (18.20) | 48.28 (17.79) | 48.27 (17.80) | 56.06 (18.36) | 56.04 (18.37) | 53.50 (17.48) | 53.50 (17.50) | 47.91 (18.04) | 47.86 (17.86) |
| | Sex, *n* (%) | | | | | | | | | | |
| | Females | 205,759 (63.73) | 48,654 (66.82) | 15,549 (38.16) | 3,111 (38.12) | 12,336 (57.43) | 2,468 (57.40) | 61,675 (70.95) | 12,341 (70.94) | 18,3790 (69.07) | 39,126 (70.17) |
| | Infection month, median (IQR) | 11 (1) | 11 (1) | 11 (2) | 11 (2) | 11 (1) | 11 (1) | 11 (1) | 11 (1) | 11 (1) | 11 (1) |
| | Infection year, median (IQR) | 2020 (0) | 2020 (0) | 2020 (0) | 2020 (0) | 2020 (0) | 2020 (0) | 2020 (0) | 2020 (0) | 2020 (0) | 2020 (0) |
| | CCI, mean (SD) | 1.32 (1.85) | 1.55 (2.03) | 1.75 (2.1) | 1.79 (2.17) | 1.98 (2.25) | 1.99 (2.26) | 1.78 (2.12) | 1.82 (2.2) | 1.33 (1.85) | 1.43 (1.93) |
| 3 | Total, *n* | 443,728 | 99,307 | 63,623 | 12,768 | 26,859 | 5,405 | 113,967 | 22,900 | 362,380 | 76,462 |
| | Age, mean (SD) | 47.35 (17.46) | 47.82 (17.23) | 44.97 (16.25) | 44.98 (16.26) | 50.51 (17.04) | 50.58 (17.04) | 51.80 (16.51) | 51.81 (16.53) | 46.72 (17.21) | 46.79 (17.04) |
| | Sex, *n* (%) | | | | | | | | | | |
| | Females | 273,385 (61.61) | 64,115 (64.56) | 22,267 (35.00) | 4,466 (34.98) | 13,875 (51.66) | 2,792 (51.66) | 79,484 (69.74) | 15,971 (69.74) | 243,249 (67.13) | 52,333 (68.44) |
| | Vaccination status, *n* (%) | | | | | | | | | | |
| | Not vaccinated | 440,565 (99.29) | 98,185 (98.87) | 63,295 (99.48) | 12,673 (99.26) | 26,394 (98.27) | 5,281 (97.71) | 112,701 (98.89) | 22,576 (98.59) | 359,796 (99.29) | 75,648 (98.94) |
| | First dose | 2,906 (0.65) | 993 (1.00) | 296 (0.47) | 80 (0.63) | 441 (1.64) | 117 (2.16) | 1,164 (1.02) | 285 (1.24) | 2,372 (0.65) | 713 (0.93) |
| | Full vaccination | 257 (0.06) | 129 (0.13) | 32 (0.05) | 15 (0.12) | 24 (0.09) | 7 (0.13) | 102 (0.09) | 39 (0.17) | 212 (0.06) | 101 (0.13) |
| | Booster | 0 (0.00) | 0 (0.00) | 0 (0.00) | 0 (0.00) | 0 (0.00) | 0 (0.00) | 0 (0.00) | 0 (0.00) | 0 (0.00) | 0 (0.00) |
| | Infection month, median (IQR) | 2 (2) | 2 (2) | 2 (2) | 2 (2) | 2 (2) | 2 (2) | 2 (2) | 2 (2) | 2 (2) | 2 (2) |
| | Infection year, median (IQR) | 2021 (0) | 2021 (0) | 2021 (0) | 2021 (0) | 2021 (0) | 2021 (0) | 2021 (0) | 2021 (0) | 2021 (0) | 2021 (0) |
| | CCI, mean (SD) | 1.22 (1.74) | 1.43 (1.89) | 1.49 (1.89) | 1.54 (2.02) | 1.59 (2.02) | 1.62 (2.07) | 1.65 (1.97) | 1.68 (2.06) | 1.24 (1.73) | 1.36 (1.82) |

(*Continued*)

**Table 1.** (Continued)

| Epoch | Characteristic | Any mental disorder | | Substance use disorders | | Psychotic disorders | | Affective disorders | | Anxiety disorders | |
|---|---|---|---|---|---|---|---|---|---|---|---|
| | | unexposed | exposed | unexposed | exposed | unexposed | exposed | unexposed | exposed | unexposed | exposed |
| 4 | Total, n | 95,357 | 22,638 | 14,480 | 3,016 | 5,334 | 1,108 | 24,092 | 5,079 | 78,317 | 17,643 |
| | Age, mean (SD) | 43.01 (17.46) | 44.12 (17.48) | 41.84 (16.16) | 42.05 (16.23) | 47.55 (16.73) | 47.80 (16.87) | 48.79 (16.27) | 49.20 (16.42) | 42.41 (17.29) | 43.03 (17.30) |
| | Sex, n (%) | | | | | | | | | | |
| | Females | 58,648 (61.50) | 14,589 (64.44) | 5,443 (37.59) | 1,131 (37.50) | 2,658 (49.83) | 551 (49.73) | 16,499 (68.48) | 3,500 (68.91) | 52,168 (66.61) | 12,104 (68.61) |
| | Vaccination status, n (%) | | | | | | | | | | |
| | Not vaccinated | 81,471 (85.44) | 18,897 (83.47) | 12,942 (89.38) | 2,660 (88.20) | 4,557 (85.43) | 934 (84.30) | 19,835 (82.33) | 4,105 (80.82) | 66,661 (85.12) | 14,707 (83.36) |
| | First dose | 2,266 (2.38) | 727 (3.21) | 215 (1.48) | 65 (2.16) | 159 (2.98) | 42 (3.79) | 696 (2.89) | 188 (3.70) | 1,866 (2.38) | 566 (3.21) |
| | Full vaccination | 11,620 (12.19) | 3,014 (13.31) | 1,323 (9.14) | 291 (9.65) | 618 (11.59) | 132 (11.91) | 3,561 (14.78) | 786 (15.48) | 9,790 (12.50) | 2,370 (13.43) |
| | Booster | 0 (0.00) | 0 (0.00) | 0 (0.00) | 0 (0.00) | 0 (0.00) | 0 (0.00) | 0 (0.00) | 0 (0.00) | 0 (0.00) | 0 (0.00) |
| | Infection month, median (IQR) | 5 (6) | 5 (6) | 4 (5) | 4 (5) | 4 (6) | 5 (6) | 5 (6) | 5 (6) | 5 (6) | 5 (6) |
| | Infection year, median (IQR) | 2021 (0) | 2021 (0) | 2021 (0) | 2021 (0) | 2021 (0) | 2021 (0) | 2021 (0) | 2021 (0) | 2021 (0) | 2021 (0) |
| | CCI, mean (SD) | 0.95 (1.5) | 1.19 (1.72) | 1.19 (1.66) | 1.3 (1.82) | 1.27 (1.77) | 1.35 (1.85) | 1.35 (1.77) | 1.46 (1.9) | 0.96 (1.48) | 1.14 (1.66) |
| 5 | Total, n | 832,235 | 187,321 | 110,334 | 22,205 | 37,958 | 7,626 | 198,047 | 39,860 | 703,868 | 150,211 |
| | Age, mean (SD) | 41.32 (16.82) | 42.22 (16.79) | 40.92 (15.65) | 40.97 (15.69) | 46.25 (16.76) | 46.30 (16.80) | 46.51 (16.03) | 46.58 (16.09) | 40.89 (16.65) | 41.30 (16.59) |
| | Sex, n (%) | | | | | | | | | | |
| | Females | 526,059 (63.21) | 123,962 (66.18) | 44,774 (40.58) | 8,989 (40.48) | 20,539 (54.11) | 4,126 (54.10) | 139,277 (70.33) | 28,043 (70.35) | 477,405 (67.83) | 104,241 (69.40) |
| | Vaccination status, n (%) | | | | | | | | | | |
| | Not vaccinated | 334,155 (40.15) | 74,090 (39.55) | 49,970 (45.29) | 10,021 (45.13) | 14,797 (38.98) | 2,962 (38.84) | 68,819 (34.75) | 13,809 (34.64) | 281,216 (39.95) | 59,521 (39.62) |
| | First dose | 9,740 (1.17) | 2,791 (1.49) | 1,990 (1.80) | 475 (2.14) | 609 (1.60) | 143 (1.88) | 2,187 (1.10) | 538 (1.35) | 8,316 (1.18) | 2,224 (1.48) |
| | Full vaccination | 372,694 (44.78) | 83,372 (44.51) | 46,014 (41.70) | 9,214 (41.50) | 17,050 (44.92) | 3,413 (44.75) | 92,887 (46.90) | 18,612 (46.69) | 316,251 (44.93) | 67,149 (44.70) |
| | Booster | 115,646 (13.90) | 27,068 (14.45) | 12,360 (11.20) | 2,495 (11.24) | 5,502 (14.49) | 1,108 (14.53) | 34,154 (17.25) | 6,901 (17.31) | 98,085 (13.94) | 21,317 (14.19) |
| | Infection month, median (IQR) | 2 (10) | 2 (10) | 2 (10) | 2 (10) | 2 (9) | 2 (9) | 2 (10) | 2 (10) | 2 (10) | 2 (10) |
| | Infection year, median (IQR) | 2022 (1) | 2022 (1) | 2022 (1) | 2022 (1) | 2022 (1) | 2022 (1) | 2022 (1) | 2022 (1) | 2022 (1) | 2022 (1) |
| | CCI, mean (SD) | 0.91 (1.43) | 1.12 (1.62) | 1.17 (1.63) | 1.21 (1.73) | 1.28 (1.74) | 1.29 (1.78) | 1.32 (1.73) | 1.34 (1.78) | 0.95 (1.43) | 1.07 (1.56) |

The results are presented as absolute numbers (n) with proportions (%), means with SD, and medians with IQRs. The time frames for epochs were: (1) 1st March 2020 to 30th September 2020 for epoch 1; (2) 1st October 2020 to 26th December 2020 for epoch 2; (3) 27th December 2020 to 31st March 2021 for epoch 3; (4) 1st April 2021 to 31st October 2021 for epoch 4; and (5) 1st November 2021 to 29th February 2022 for epoch 5. "Exposed" and "unexposed" refer to people with the respective mental disorder and their matched counterparts without that mental disorder, respectively. The International Classification of Diseases 10th Revision (ICD-10) diagnostic codes were (1) F10-F19, F20-F29, F30-F39, F40-F48 for any mental disorder; (2) F10-F19 for substance use disorders; (3) F20-F29 for psychotic disorders; (4) F30-F39 for affective disorders; and (5) F40-F48 for anxiety disorders.

CCI, Charlson Comorbidity Index; ICD-10, International Classification of Diseases 10th Revision; IQR, interquartile range; SD, standard deviation.

**Table 2. Descriptive statistics per matching variables, cases ascertained by diagnosis per the ICD-10 diagnostic codes coupled with prescription for psychopharmaceuticals per the ATC classification codes.**

| Epoch | Characteristic | Any mental disorder | | Substance use disorders | | Psychotic disorders | | Affective disorders | | Anxiety disorders | |
|---|---|---|---|---|---|---|---|---|---|---|---|
| | | unexposed | exposed | unexposed | exposed | unexposed | exposed | unexposed | exposed | unexposed | exposed |
| 1 | Total, *n* | 18,842 | 5,121 | 1,870 | 434 | 1,081 | 247 | 6,254 | 1,470 | 15,963 | 4,127 |
| | Age, mean (SD) | 43.02 (15.68) | 46.58 (16.83) | 42.97 (16.04) | 45.54 (17.50) | 47.16 (17.70) | 49.55 (18.88) | 46.26 (15.84) | 48.53 (16.96) | 42.81 (15.54) | 45.75 (16.59) |
| | Sex, *n* (%) | | | | | | | | | | |
| | Females | 11,483 (60.94) | 3,390 (66.20) | 822 (43.96) | 193 (44.47) | 621 (57.45) | 144 (58.30) | 4,027 (64.39) | 970 (65.99) | 10,227 (64.07) | 2,826 (68.48) |
| | Infection month, median (IQR) | 9 (1) | 9 (1) | 9 (1) | 9 (2) | 9 (1) | 9 (1) | 9 (0) | 9 (1) | 9 (1) | 9 (1) |
| | Infection year, median (IQR) | 2020 (0) | 2020 (0) | 2020 (0) | 2020 (0) | 2020 (0) | 2020 (0) | 2020 (0) | 2020 (0) | 2020 (0) | 2020 (0) |
| | CCI, mean (SD) | 0.82 (1.34) | 1.26 (1.76) | 1.16 (1.64) | 1.58 (2.16) | 1.19 (1.78) | 1.48 (1.98) | 1.08 (1.54) | 1.44 (1.9) | 0.83 (1.33) | 1.22 (1.72) |
| 2 | Total, *n* | 213,314 | 57,301 | 27,022 | 5,441 | 19,500 | 3,914 | 76,430 | 16,200 | 180,941 | 44,226 |
| | Age, mean (SD) | 48.03 (16.56) | 51.68 (17.59) | 50.83 (17.40) | 50.81 (17.41) | 55.94 (18.31) | 55.97 (18.39) | 53.00 (16.80) | 54.15 (17.36) | 47.70 (16.57) | 50.16 (17.39) |
| | Sex, *n* (%) | | | | | | | | | | |
| | Females | 131,715 (61.75) | 39,534 (68.99) | 11,562 (42.79) | 2,319 (42.62) | 11,318 (58.04) | 2,273 (58.07) | 53,282 (69.71) | 11,555 (71.33) | 120,407 (66.54) | 31,760 (71.81) |
| | Infection month, median (IQR) | 11 (1) | 11 (1) | 11 (2) | 11 (2) | 11 (1) | 11 (1) | 11 (1) | 11 (1) | 11 (1) | 11 (1) |
| | Infection year, median (IQR) | 2020 (0) | 2020 (0) | 2020 (0) | 2020 (0) | 2020 (0) | 2020 (0) | 2020 (0) | 2020 (0) | 2020 (0) | 2020 (0) |
| | CCI, mean (SD) | 1.1 (1.59) | 1.68 (2.09) | 1.99 (2.12) | 2.1 (2.32) | 1.97 (2.16) | 2.03 (2.27) | 1.62 (1.91) | 1.87 (2.22) | 1.14 (1.61) | 1.57 (2.02) |
| 3 | Total, *n* | 300,335 | 76,731 | 39,933 | 8,048 | 24,564 | 4,968 | 101,592 | 21,165 | 252,244 | 59,918 |
| | Age, mean (SD) | 47.58 (16.41) | 50.01 (16.68) | 46.65 (15.95) | 46.66 (15.97) | 50.39 (16.96) | 50.54 (17.03) | 52.01 (16.29) | 52.41 (16.39) | 47.32 (16.33) | 48.95 (16.48) |
| | Sex, *n* (%) | | | | | | | | | | |
| | Females | 181,621 (60.47) | 51,538 (67.17) | 15,807 (39.58) | 3,178 (39.49) | 12,961 (52.76) | 2,627 (52.88) | 70,433 (69.33) | 14,888 (70.34) | 164,964 (65.40) | 41,981 (70.06) |
| | Vaccination status, *n* (%) | | | | | | | | | | |
| | Not vaccinated | 298,691 (99.45) | 75,921 (98.94) | 39,746 (99.53) | 7,986 (99.23) | 24,274 (98.82) | 4,865 (97.93) | 100,860 (99.28) | 20,903 (98.76) | 250,901 (99.47) | 59,323 (99.01) |
| | First dose | 1,521 (0.51) | 732 (0.95) | 177 (0.44) | 54 (0.67) | 280 (1.14) | 96 (1.93) | 680 (0.67) | 235 (1.11) | 1,242 (0.49) | 535 (0.89) |
| | Full vaccination | 123 (0.04) | 78 (0.10) | 10 (0.03) | 8 (0.10) | 10 (0.04) | 7 (0.14) | 52 (0.05) | 27 (0.13) | 101 (0.04) | 60 (0.10) |
| | Booster | 0 (0.00) | 0 (0.00) | 0 (0.00) | 0 (0.00) | 0 (0.00) | 0 (0.00) | 0 (0.00) | 0 (0.00) | 0 (0.00) | 0 (0.00) |
| | Infection month, median (IQR) | 2 (2) | 2 (2) | 2 (2) | 2 (2) | 2 (2) | 2 (2) | 2 (2) | 2 (2) | 2 (2) | 2 (2) |
| | Infection year, median (IQR) | 2021 (0) | 2021 (0) | 2021 (0) | 2021 (0) | 2021 (0) | 2021 (0) | 2021 (0) | 2021 (0) | 2021 (0) | 2021 (0) |
| | CCI, mean (SD) | 1.1 (1.58) | 1.56 (1.97) | 1.7 (1.93) | 1.81 (2.17) | 1.56 (1.92) | 1.63 (2.06) | 1.58 (1.87) | 1.74 (2.08) | 1.13 (1.58) | 1.49 (1.89) |

(*Continued*)

**Table 2.** (Continued)

| Epoch | Characteristic | Any mental disorder | | Substance use disorders | | Psychotic disorders | | Affective disorders | | Anxiety disorders | |
|---|---|---|---|---|---|---|---|---|---|---|---|
| | | unexposed | exposed | unexposed | exposed | unexposed | exposed | unexposed | exposed | unexposed | exposed |
| 4 | Total, *n* | 61,983 | 16,322 | 8,311 | 1,795 | 4,723 | 1,006 | 20,439 | 4,555 | 51,628 | 12,870 |
| | Age, mean (SD) | 44.20 (16.12) | 46.75 (16.60) | 44.08 (15.48) | 44.55 (15.81) | 47.03 (16.33) | 47.48 (16.55) | 48.59 (15.72) | 49.68 (16.04) | 43.75 (16.05) | 45.69 (16.42) |
| | Sex, *n* (%) | | | | | | | | | | |
| | Females | 38,225 (61.67) | 10,925 (66.93) | 3,494 (42.04) | 763 (42.51) | 2,374 (50.26) | 513 (50.99) | 13,863 (67.83) | 3,158 (69.33) | 34,093 (66.04) | 9,026 (70.13) |
| | Vaccination status, *n* (%) | | | | | | | | | | |
| | Not vaccinated | 52,930 (85.39) | 13,489 (82.64) | 7,465 (89.82) | 1,583 (88.19) | 4,143 (87.72) | 862 (85.69) | 17,007 (83.21) | 3,695 (81.12) | 43,931 (85.09) | 10,620 (82.52) |
| | First dose | 1,371 (2.21) | 524 (3.21) | 91 (1.09) | 34 (1.89) | 106 (2.24) | 33 (3.28) | 501 (2.45) | 160 (3.51) | 1,129 (2.19) | 407 (3.16) |
| | Full vaccination | 7,682 (12.39) | 2,309 (14.15) | 755 (9.08) | 178 (9.92) | 474 (10.04) | 111 (11.03) | 2,931 (14.34) | 700 (15.37) | 6,568 (12.72) | 1,843 (14.32) |
| | Booster | 0 (0.00) | 0 (0.00) | 0 (0.00) | 0 (0.00) | 0 (0.00) | 0 (0.00) | 0 (0.00) | 0 (0.00) | 0 (0.00) | 0 (0.00) |
| | Infection month, median (IQR) | 5 (6) | 5 (6) | 4 (5) | 4 (5) | 4 (6) | 4 (6) | 5 (6) | 5 (6) | 5 (6) | 5 (6) |
| | Infection year, median (IQR) | 2021 (0) | 2021 (0) | 2021 (0) | 2021 (0) | 2021 (0) | 2021 (0) | 2021 (0) | 2021 (0) | 2021 (0) | 2021 (0) |
| | CCI, mean (SD) | 0.89 (1.38) | 1.29 (1.76) | 1.31 (1.63) | 1.52 (1.91) | 1.17 (1.63) | 1.31 (1.81) | 1.23 (1.63) | 1.46 (1.89) | 0.9 (1.37) | 1.25 (1.73) |
| 5 | Total, *n* | 544,077 | 135,003 | 65,572 | 13,286 | 34,590 | 6,981 | 173,299 | 36,134 | 467,140 | 109,359 |
| | Age, mean (SD) | 42.51 (15.60) | 44.98 (16.12) | 43.18 (15.32) | 43.28 (15.40) | 46.08 (16.45) | 46.19 (16.55) | 46.74 (15.68) | 47.30 (15.88) | 42.32 (15.55) | 44.11 (15.92) |
| | Sex, *n* (%) | | | | | | | | | | |
| | Females | 345,333 (63.47) | 92,640 (68.62) | 29,877 (45.56) | 6,038 (45.45) | 18,999 (54.93) | 3,843 (55.05) | 12,1041 (69.85) | 25,577 (70.78) | 314,398 (67.30) | 77,820 (71.16) |
| | Vaccination status, *n* (%) | | | | | | | | | | |
| | Not vaccinated | 207,757 (38.19) | 49,441 (36.62) | 27,772 (42.35) | 5,623 (42.32) | 13,325 (38.52) | 2,675 (38.32) | 59,141 (34.13) | 12,226 (33.84) | 17,7201 (37.93) | 40,260 (36.81) |
| | First dose | 6,037 (1.11) | 1,777 (1.32) | 1,032 (1.57) | 257 (1.93) | 484 (1.40) | 122 (1.75) | 1,601 (0.92) | 429 (1.19) | 5,213 (1.12) | 1,463 (1.34) |
| | Full vaccination | 25,0276 (46.00) | 61,676 (45.68) | 28,349 (43.23) | 5,696 (42.87) | 15,708 (45.41) | 3,152 (45.15) | 822,41 (47.46) | 16,992 (47.02) | 215,799 (46.20) | 50,102 (45.81) |
| | Booster | 80,007 (14.71) | 22,109 (16.38) | 8,419 (12.84) | 1,710 (12.87) | 5,073 (14.67) | 1,032 (14.78) | 30,316 (17.49) | 6,487 (17.95) | 68,927 (14.76) | 17,534 (16.03) |
| | Infection month, median (IQR) | 2 (10) | 2 (10) | 2 (10) | 2 (10) | 2 (9) | 2 (9) | 2 (10) | 2 (10) | 2 (10) | 2 (10) |
| | Infection year, median (IQR) | 2022 (1) | 2022 (1) | 2022 (1) | 2022 (1) | 2022 (1) | 2022 (1) | 2022 (1) | 2022 (1) | 2022 (1) | 2022 (1) |
| | CCI, mean (SD) | 0.87 (1.32) | 1.26 (1.7) | 1.38 (1.72) | 1.46 (1.88) | 1.27 (1.71) | 1.3 (1.79) | 1.25 (1.62) | 1.39 (1.8) | 0.9 (1.34) | 1.21 (1.65) |

The results are presented as absolute numbers (*n*) with proportions (%), means with SDs, and medians with IQRs. The time frames for epochs were: (1) 1st March 2020 to 30th September 2020 for epoch 1; (2) 1st October 2020 to 26th December 2020 for epoch 2; (3) 27th December 2020 to 31st March 2021 for epoch 3; (4) 1st April 2021 to 31st October 2021 for epoch 4; and (5) 1st November 2021 to 29th February 2022 for epoch 5. "Exposed" and "unexposed" refer to people with the respective mental disorder and their matched counterparts without that mental disorder, respectively. The International Classification of Diseases 10th Revision (ICD-10) diagnostic codes were (1) F10-F19, F20-F29, F30-F39, F40-F48 for any mental disorder; (2) F10-F19 for substance use disorders; (3) F20-F29 for psychotic disorders; (4) F30-F39 for affective disorders; and (5) F40-F48 for anxiety disorders. The considered psychopharmaceuticals per the Anatomical Therapeutic Chemical (ATC) classification codes were (1) anxiolytics/hypnotics/sedatives (N05B, N05C); (2) antidepressants (N06A); (3) antipsychotics (N05A); and (4) stimulants (N06B).

ATC, Anatomical Therapeutic Chemical; CCI, Charlson Comorbidity Index; ICD-10, International Classification of Diseases 10th Revision; IQR, interquartile range; SD, standard deviation.

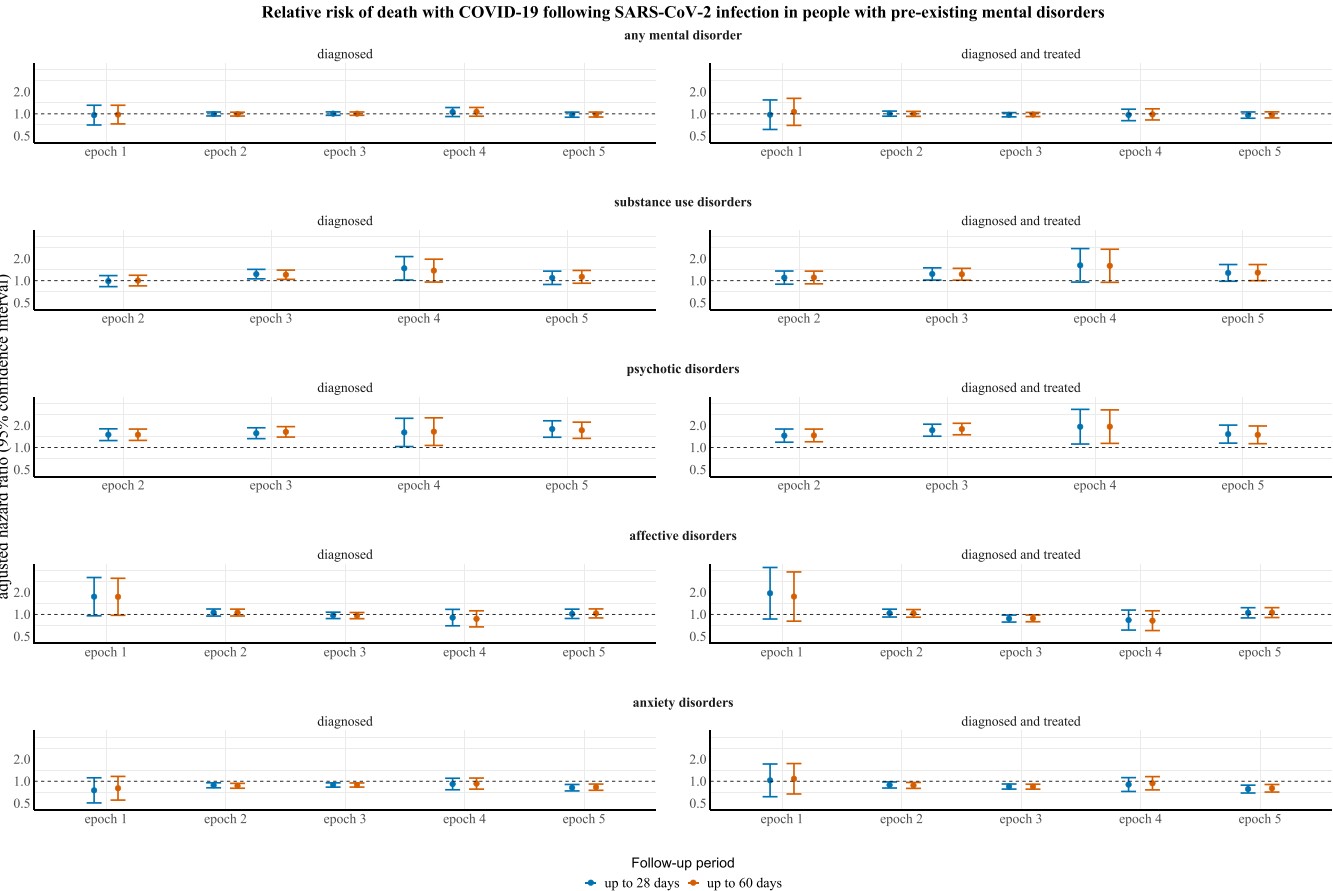

**Fig 1. Relative risk of death with COVID-19 following SARS-CoV-2 infection in people with preexisting mental disorders.** All results are expressed as HRs with 95% CIs. The models were adjusted for matching variables and all additional confounders. The time frames for epochs were: (1) 1st March 2020 to 30th September 2020 for epoch 1; (2) 1st October 2020 to 26th December 2020 for epoch 2; (3) 27th December 2020 to 31st March 2021 for epoch 3; (4) 1st April 2021 to 31st October 2021 for epoch 4; and (5) 1st November 2021 to 29th February 2022 for epoch 5. "Diagnosed" refers to cases ascertained by diagnosis per the International Classification of Diseases 10th Revision (ICD-10) diagnostic codes: (1) F10-F19, F20-F29, F30-F39, F40-F48 for any mental disorder; (2) F10-F19 for substance use disorders; (3) F20-F29 for psychotic disorders; (4) F30-F39 for affective disorders; and (5) F40-F48 for anxiety disorders. "Diagnosed and treated" refers to cases ascertained by diagnosis per the above ICD-10 codes coupled with prescription for (1) anxiolytics/hypnotics/ sedatives (N05B, N05C); (2) antidepressants (N06A); (3) antipsychotics (N05A); or (4) stimulants (N06B) per the ATC classification codes. The models could not be reliably fit for substance use and psychotic disorders in epoch 1. ATC, Anatomical Therapeutic Chemical; CI, confidence interval; COVID-19, Coronavirus Disease 2019; HR, hazard ratio; SARS-CoV-2, Severe Acute Respiratory Syndrome Coronavirus 2.

consistent results, with decreased risks of all-cause mortality in epoch 2, 3, and 5, both 28 and 60 days postinfection. The remaining models for anxiety disorders were consistent with a null effect. In addition, people diagnosed and treated for affective disorders had a decreased risk of all-cause death in epoch 3, both 28 and 60 following SARS-CoV-2 infection, but all other models involving affective disorders were broadly consistent with a null effect. The results for any studied mental disorder were—regardless of case ascertainment definition—consistent with a null effect in all epochs. For detailed results, see Figs 2 and S102–201 in S1 File and S11–S13 Tables.

## Sensitivity analyses

In negative control exposure analyses, the proportion of non-null tests did not exceed 15% for substance use disorders, 15% for psychotic disorders, 20% for affective disorders, 40% for

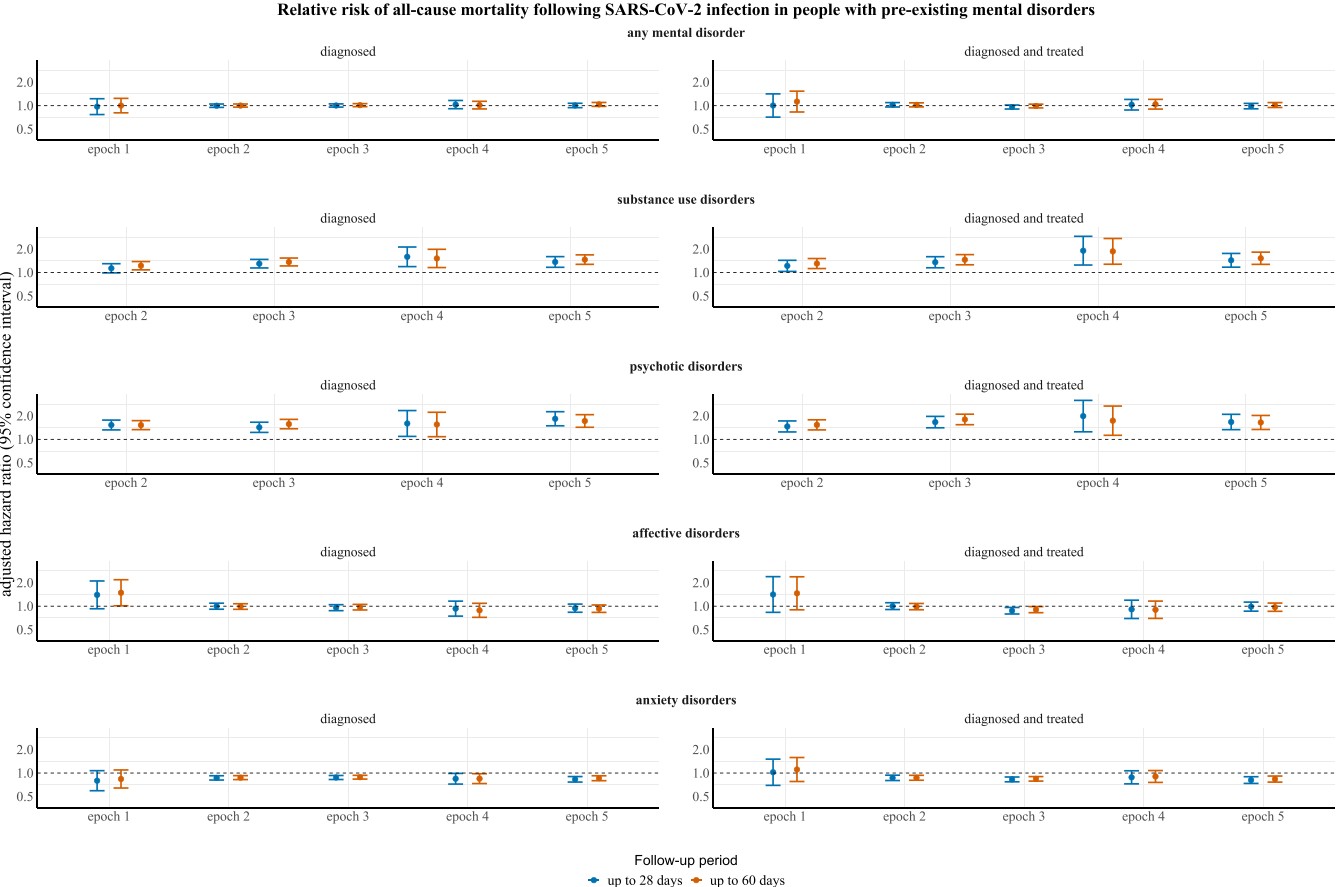

**Fig 2. Relative risk of all-cause mortality following SARS-CoV-2 infection in people with preexisting mental disorders.** All results are expressed as HRs with 95% CIs. The models were adjusted for matching variables and all additional confounders. The time frames for epochs were: (1) 1st March 2020 to 30th September 2020 for epoch 1; (2) 1st October 2020 to 26th December 2020 for epoch 2; (3) 27th December 2020 to 31st March 2021 for epoch 3; (4) 1st April 2021 to 31st October 2021 for epoch 4; and (5) 1st November 2021 to 29th February 2022 for epoch 5. "Diagnosed" refers to cases ascertained by diagnosis per the International Classification of Diseases 10th Revision (ICD-10) diagnostic codes: (1) F10-F19, F20-F29, F30-F39, F40-F48 for any mental disorder; (2) F10-F19 for substance use disorders; (3) F20-F29 for psychotic disorders; (4) F30-F39 for affective disorders; and (5) F40-F48 for anxiety disorders. "Diagnosed and treated" refers to cases ascertained by diagnosis per the above ICD-10 codes coupled with prescription for (1) anxiolytics/hypnotics/sedatives (N05B, N05C); (2) antidepressants (N06A); (3) antipsychotics (N05A); or (4) stimulants (N06B) per the ATC classification codes. The models could not be reliably fit for substance use and psychotic disorders in epoch 1. ATC, Anatomical Therapeutic Chemical; CI, confidence interval; HR, hazard ratio; SARS-CoV-2, Severe Acute Respiratory Syndrome Coronavirus 2.

anxiety disorders, and 45% for any of the studied mental disorders. Proportion of non-null tests closer to 0% increases the confidence in the lack of collider bias. See details, including averaged HRs, in S14 Table.

For models not consistent with a null-effect, the *E*-values ranged from 1.71 to 3.22 for substance use disorders, from 2.21 to 3.39 for psychotic disorders, from 1.50 to 1.88 for anxiety disorders, and from 1.45 to 1.55 for affective disorders. *E*-values further away from 1 increase the confidence that the detected associations are not due to unaccounted for confounding. See details in S15 and S16 Tables.

## Discussion

Using Czech national health register data, we demonstrated that people with preexisting psychotic disorders were more likely to die with COVID-19 or due to any cause following

SARS-CoV-2 infection throughout the pandemic. We demonstrated less robust associations for deaths with COVID-19 in people with substance use disorders but they had a consistent and sustained elevated risk of all-cause mortality following SARS-CoV-2 infection. The 2 exposure definitions produced broadly consistent results across each epoch-mental disorder combination. This detected excess mortality is not fully explicable by differences in vaccination uptake or clinically recorded physical comorbidities between people with and without preexisting substance use and psychotic disorders. Separately, people with anxiety disorders demonstrated decreased risk of death with COVID-19 and from all-causes in multiple epochs, whereas the risk in people with affective disorders was broadly consistent with a null effect throughout the pandemic.

Our findings are in line with existing evidence showing elevated mortality risk in people with psychotic [10–17] and substance use disorders [18–21]; however, we demonstrated that these health disparities were consistent throughout the pandemic and persisted at its later phases too. Robust control for vaccination uptake or clinically recorded somatic comorbidity in our study did not reverse the increased risk of death.

These results, broadly consistent with a citywide study from the United Kingdom [45], demonstrate vulnerability in these patient groups that cannot be fully explained by differences in vaccination uptake or clinically recorded physical comorbidities. This suggest that other individual and structural factors might be responsible for the detected outcomes. Inequalities in access to healthcare and differences in the quality of care received cannot be discounted as contributing to the excess mortality. Previous studies showed that these patient groups may face delayed diagnosis [46,47] if physical health conditions are recognized at all [48]; such suboptimal episodes of care may be related to, among other things, incorrectly attributing the symptoms of somatic conditions to mental disorders [49]. Thus, people with psychotic and substance use disorders potentially have more severe and insufficiently addressed or clinically unrecognized physical comorbidities that contributed to lower survival.

People with lower socioeconomic status were less likely to be tested for SARS-CoV-2 infection [50–52], but more likely to experience delayed test results [53]. Both substance use and psychotic disorders are negatively associated with socioeconomic status [54–57]. It is plausible that SARS-CoV-2 infection in these people was recognized comparatively late and that this potentially adversely influenced the therapeutic response.

Further, negative health behaviors such as smoking tobacco and suboptimal nutrition and physical activity are common in these patient groups [58,59] and may have contributed to worse prognosis. Pharmacological treatments for psychotic disorders are known to contribute to metabolic disturbances [60], interact with or limit the use of treatment for somatic conditions, thus potentially contributing to lower survival also post-SARS-CoV-2 infection. However, pre-pandemic research has demonstrated that the use of antipsychotics is associated with decreased risk of all-cause mortality in people with schizophrenia [61], with no differences between concomitant use of several ones compared with monotherapy [62]. Alternatively, lower adherence to prescription medications [63] that would contribute to worsened overall health at the baseline, cannot be ruled out.

Overall, the existing evidence and our results suggest the presence of fatal but largely tractable health inequalities in these patient groups. Wholesale system approaches, predicated on evidence-based policy changes and, ideally, combined with evaluation, are required to address the multilayered factors behind these fatal health inequalities. In a future pandemic or other health emergency, substance use and psychotic disorders need to be considered as a specific vulnerability factor beyond liability to the health threat, itself.

Regarding other psychiatric conditions, our study further showed decreased mortality risk in people with anxiety disorders in multiple epochs, contrasting findings of existing studies

that demonstrated elevated risks [11,13,16,22] or null effects [15,18]. Multiple factors might be responsible for these differences, including the scope of data, definition of cases and the comparison groups, as well as analytical approaches. In 2 French studies, for example, anxiety disorders were identified through a hospital register [18,22], while our study included cases in all healthcare settings, likely capturing less severe cases with better outcomes. In a UK Biobank study, narrower definitions of anxiety- and stress-related disorders were used [13], focusing on more severe conditions. The study also did not use a matched-cohort design [13], making it difficult to rule out that adjusting for clinical, sociodemographic, and behavioral confounders in regression models still led to residual confounding due to covariate imbalance. Similarly, in a QResearch database study, the authors did not implement a matched-cohort design and also did not condition on positive test for a SARS-CoV-2 infection, hence, comparisons involved different populations [16].

Evidence shows that anxiety symptoms may have been linked to higher endorsement of preventive measures by heightened contamination fear [64,65]. Together with our findings of decreased mortality risk, this may suggest that excessive preoccupation with COVID-19–related events may have facilitated early detection and improved the health outcomes in some people with anxiety disorders.

The broadly null effect we detected for mortality in people with affective disorders was reported before also elsewhere [15]; however, other studies demonstrated an elevated risk [11,14,16,23]. As with anxiety disorders, the factors responsible for these diverging results are most likely multiple and include breadth of data (all healthcare settings versus specific settings), case definitions (all affective disorders diagnostic codes versus specific diagnoses of affective disorders), and analytical choices (matched-cohorts versus unmatched-cohorts, conditioning on positive SARS-CoV-2 infection tests versus not conditioning on these).

Strengths included the use of national, whole population, fully standardized data on SARS-CoV-2 infection status, healthcare utilization, and mortality. Next, we investigated the robustness of our results through using multiple definitions of the exposure and the outcome. Then, we used negative control exposures to explore the potential presence of collider bias and $E$-values to establish what level of unaccounted for confounding would explain away the observed associations.

This study has also some limitations. First, we used broad diagnostic categories of mental disorders; thus disregarding the diagnostic heterogeneity in these, including the differing levels of severity (e.g., bipolar disorder in affective disorders). Second, we matched on key sociodemographic and clinical covariates and subsequently adjusted for a wide range of additional health-related confounders; however, we did not control for a number of previously identified influential clinical, sociodemographic, and behavioral confounders [66]. In particular, we had no information on body mass index, smoking status, and socioeconomic status per se; however, these are likely to influence an individual's overall health [67–72], which we considered by controlling for a comorbidity index, overall number of contacts with inpatient and outpatient services, and prescription medications, including antihypertensives and statins that would be administered for conditions commonly present in people who smoke or who are obese [73–76]. These steps likely reduced the level of confounding due to these covariates; however, we cannot rule out that part of the detected associations is still due to residual confounding, both related to these known and directly unmeasured and potential unknown confounders, with the most plausible direction of bias being the overestimation of true effects. Third, we were able to match the vast majority of people with mental disorders with counterparts without mental disorders; however, "bias due to incomplete matching" cannot be ruled out [77]. In particular, the unmatched individuals with mental disorders tended to be, on average, older and have a higher number of comorbidities. Since these individuals can be expected

to have the worse outcomes postinfection, the most plausible direction of bias seems to be the underestimation of true effects. Fourth, while clinically recorded physical comorbidities are among the factors most strongly associated with worse prognosis following a SARS-CoV-2 infection [78], we cannot rule out that in some individuals with mental disorders, they would act as a mediator instead of a confounder, thus raising the possibility of overadjustment bias [79]. Fifth, both the curator of the data, IHIS, and the insurance companies who use them to reimburse service providers employ mechanisms to ensure the validity of data; however, all diagnoses used in this study have not been fully validated yet. Thus, under-registration and/or errors in diagnoses coding cannot be ruled out. Sixth, some of our analyses included considerably few individuals, leading to profound uncertainty in our estimates. Seventh, we did not investigate survival following SARS-CoV-2 re-infections in people with preexisting disorders. Eighth, we did not investigate the responses to medications and/or other treatment modalities following the infection itself in people with preexisting mental disorders. Ninth, we did not consider the outcomes of people with multiple psychiatric conditions. Last, the follow-up period following infection was short; however, we had no information on emigration status, and we cannot rule out that some individuals were lost to follow-up.

People with preexisting psychotic, and, less robustly, substance use disorders demonstrated persistently lower survival following SARS-CoV-2 infection throughout the pandemic. While we cannot rule out that part of the detected associations is due to residual confounding, the consistently increased vulnerability beyond vaccination uptake and clinically recorded physical comorbidity aligns with existing evidence on fatal health inequalities in these patient groups and underlines the importance of implementing systemic efforts to fully reverse these. To at least reduce these disparities, it must be assured that these patient groups are included in future vaccination campaigns.

## Supporting information

**S1 Methods. Supplementary methods.**
(DOCX)

**S1 RECORD Checklist. The REporting of studies Conducted using Observational Routinely-collected health Data (RECORD) Checklist.**
(DOCX)

**S1 Table. Unmatched individuals per cohorts.**
(DOCX)

**S2 Table. Characteristics of unmatched individuals, cases ascertained by diagnosis per the International Classification of Diseases 10th Revision (ICD-10) diagnostic codes.**
(DOCX)

**S3 Table. Characteristics of unmatched individuals, cases ascertained by diagnosis per the International Classification of Diseases 10th Revision (ICD-10) diagnostic codes coupled with prescription for psychopharmaceuticals per the Anatomical Therapeutic Chemical (ATC) classification codes.**
(DOCX)

**S4 Table. Number of matches for cases ascertained by diagnosis per the International Classification of Diseases 10th Revision (ICD-10) diagnostic codes.**
(DOCX)

**S5 Table. Number of matches for cases ascertained by diagnosis per the International Classification of Diseases 10th Revision (ICD-10) diagnostic codes coupled with prescription for psychopharmaceuticals per the Anatomical Therapeutic Chemical (ATC) classification codes.**
(DOCX)

**S6 Table. Descriptive statistics on additional confounders for cases ascertained by diagnosis per the International Classification of Diseases 10th Revision (ICD-10) diagnostic codes.**
(DOCX)

**S7 Table. Descriptive statistics on additional confounders for cases ascertained by diagnosis per the International Classification of Diseases 10th Revision (ICD-10) diagnostic codes coupled with prescription for psychopharmaceuticals per the Anatomical Therapeutic Chemical (ATC) classification codes.**
(DOCX)

**S8 Table. Risk of death with COVID-19 up to 28 days in people with preexisting mental disorders.**
(DOCX)

**S9 Table. Risk of death with COVID-19 up to 60 days in people with preexisting mental disorders.**
(DOCX)

**S10 Table. Absolute risk of death with COVID-19 in people with preexisting mental disorders.**
(DOCX)

**S11 Table. Risk of all-cause mortality up to 28 days in people with preexisting mental disorders.**
(DOCX)

**S12 Table. Risk of all-cause mortality up to 60 days in people with preexisting mental disorders.**
(DOCX)

**S13 Table. Absolute risk of all-cause mortality in people with preexisting mental disorders.**
(DOCX)

**S14 Table. Negative control exposure analyses.**
(DOCX)

**S15 Table. E-values for models on deaths with COVID-19.**
(DOCX)

**S16 Table. E-values for models on all-cause mortality.**
(DOCX)

**S1 File.** S1 Fig. Directed acyclic graph. S2 Fig. Cumulative probability with 95% confidence interval of death with COVID-19 up to 28 days in people with any mental disorder in epoch 1, cases ascertained by diagnosis per the International Classification of Diseases 10th Revision (ICD-10) diagnostic codes. **S3 Fig.** Cumulative probability with 95% confidence interval of death with COVID-19 up to 28 days in people with any mental disorder in epoch 2, cases ascertained by diagnosis per the International Classification of Diseases 10th Revision (ICD-

10) diagnostic codes. **S4 Fig.** Cumulative probability with 95% confidence interval of death with COVID-19 up to 28 days in people with any mental disorder in epoch 3, cases ascertained by diagnosis per the International Classification of Diseases 10th Revision (ICD-10) diagnostic codes. **S5 Fig.** Cumulative probability with 95% confidence interval of death with COVID-19 up to 28 days in people with any mental disorder in epoch 4, cases ascertained by diagnosis per the International Classification of Diseases 10th Revision (ICD-10) diagnostic codes. **S6 Fig.** Cumulative probability with 95% confidence interval of death with COVID-19 up to 28 days in people with any mental disorder in epoch 5, cases ascertained by diagnosis per the International Classification of Diseases 10th Revision (ICD-10) diagnostic codes. **S7 Fig.** Cumulative probability with 95% confidence interval of death with COVID-19 up to 28 days in people with any mental disorder in epoch 1, cases ascertained by diagnosis per the International Classification of Diseases 10th Revision (ICD-10) diagnostic codes coupled with prescription for psychopharmaceuticals per the Anatomical Therapeutic Chemical (ATC) classification codes. **S8 Fig.** Cumulative probability with 95% confidence interval of death with COVID-19 up to 28 days in people with any mental disorder in epoch 2, cases ascertained by diagnosis per the International Classification of Diseases 10th Revision (ICD-10) diagnostic codes coupled with prescription for psychopharmaceuticals per the Anatomical Therapeutic Chemical (ATC) classification codes. **S9 Fig.** Cumulative probability with 95% confidence interval of death with COVID-19 up to 28 days in people with any mental disorder in epoch 3, cases ascertained by diagnosis per the International Classification of Diseases 10th Revision (ICD-10) diagnostic codes coupled with prescription for psychopharmaceuticals per the Anatomical Therapeutic Chemical (ATC) classification codes. **S10 Fig.** Cumulative probability with 95% confidence interval of death with COVID-19 up to 28 days in people with any mental disorder in epoch 4, cases ascertained by diagnosis per the International Classification of Diseases 10th Revision (ICD-10) diagnostic codes coupled with prescription for psychopharmaceuticals per the Anatomical Therapeutic Chemical (ATC) classification codes. **S11 Fig.** Cumulative probability with 95% confidence interval of death with COVID-19 up to 28 days in people with any mental disorder in epoch 5, cases ascertained by diagnosis per the International Classification of Diseases 10th Revision (ICD-10) diagnostic codes coupled with prescription for psychopharmaceuticals per the Anatomical Therapeutic Chemical (ATC) classification codes. **S12 Fig.** Cumulative probability with 95% confidence interval of death with COVID-19 up to 28 days in people with substance use disorders in epoch 1, cases ascertained by diagnosis per the International Classification of Diseases 10th Revision (ICD-10) diagnostic codes. **S13 Fig.** Cumulative probability with 95% confidence interval of death with COVID-19 up to 28 days in people with substance use disorders in epoch 2, cases ascertained by diagnosis per the International Classification of Diseases 10th Revision (ICD-10) diagnostic codes. **S14 Fig.** Cumulative probability with 95% confidence interval of death with COVID-19 up to 28 days in people with substance use disorders in epoch 3, cases ascertained by diagnosis per the International Classification of Diseases 10th Revision (ICD-10) diagnostic codes. **S15 Fig.** Cumulative probability with 95% confidence interval of death with COVID-19 up to 28 days in people with substance use disorders in epoch 4, cases ascertained by diagnosis per the International Classification of Diseases 10th Revision (ICD-10) diagnostic codes. **S16 Fig.** Cumulative probability with 95% confidence interval of death with COVID-19 up to 28 days in people with substance use disorders in epoch 5, cases ascertained by diagnosis per the International Classification of Diseases 10th Revision (ICD-10) diagnostic codes. **S17 Fig.** Cumulative probability with 95% confidence interval of death with COVID-19 up to 28 days in people with substance use disorders in epoch 1, cases ascertained by diagnosis per the International Classification of Diseases 10th Revision (ICD-10) diagnostic codes coupled with prescription for psychopharmaceuticals per the Anatomical Therapeutic Chemical (ATC) classification

codes. **S18 Fig.** Cumulative probability with 95% confidence interval of death with COVID-19 up to 28 days in people with substance use disorders in epoch 2, cases ascertained by diagnosis per the International Classification of Diseases 10th Revision (ICD-10) diagnostic codes coupled with prescription for psychopharmaceuticals per the Anatomical Therapeutic Chemical (ATC) classification codes. **S19 Fig.** Cumulative probability with 95% confidence interval of death with COVID-19 up to 28 days in people with substance use disorders in epoch 3, cases ascertained by diagnosis per the International Classification of Diseases 10th Revision (ICD-10) diagnostic codes coupled with prescription for psychopharmaceuticals per the Anatomical Therapeutic Chemical (ATC) classification codes. **S20 Fig.** Cumulative probability with 95% confidence interval of death with COVID-19 up to 28 days in people with substance use disorders in epoch 4, cases ascertained by diagnosis per the International Classification of Diseases 10th Revision (ICD-10) diagnostic codes coupled with prescription for psychopharmaceuticals per the Anatomical Therapeutic Chemical (ATC) classification codes. **S21 Fig.** Cumulative probability with 95% confidence interval of death with COVID-19 up to 28 days in people with substance use disorders in epoch 5, cases ascertained by diagnosis per the International Classification of Diseases 10th Revision (ICD-10) diagnostic codes coupled with prescription for psychopharmaceuticals per the Anatomical Therapeutic Chemical (ATC) classification codes. **S22 Fig.** Cumulative probability with 95% confidence interval of death with COVID-19 up to 28 days in people with psychotic disorders in epoch 1, cases ascertained by diagnosis per the International Classification of Diseases 10th Revision (ICD-10) diagnostic codes. **S23 Fig.** Cumulative probability with 95% confidence interval of death with COVID-19 up to 28 days in people with psychotic disorders in epoch 2, cases ascertained by diagnosis per the International Classification of Diseases 10th Revision (ICD-10) diagnostic codes. **S24 Fig.** Cumulative probability with 95% confidence interval of death with COVID-19 up to 28 days in people with psychotic disorders in epoch 3, cases ascertained by diagnosis per the International Classification of Diseases 10th Revision (ICD-10) diagnostic codes. **S25 Fig.** Cumulative probability with 95% confidence interval of death with COVID-19 up to 28 days in people with psychotic disorders in epoch 4, cases ascertained by diagnosis per the International Classification of Diseases 10th Revision (ICD-10) diagnostic codes. **S26 Fig.** Cumulative probability with 95% confidence interval of death with COVID-19 up to 28 days in people with psychotic disorders in epoch 5, cases ascertained by diagnosis per the International Classification of Diseases 10th Revision (ICD-10) diagnostic codes. **S27 Fig.** Cumulative probability with 95% confidence interval of death with COVID-19 up to 28 days in people with psychotic disorders in epoch 1, cases ascertained by diagnosis per the International Classification of Diseases 10th Revision (ICD-10) diagnostic codes coupled with prescription for psychopharmaceuticals per the Anatomical Therapeutic Chemical (ATC) classification codes. **S28 Fig.** Cumulative probability with 95% confidence interval of death with COVID-19 up to 28 days in people with psychotic disorders in epoch 2, cases ascertained by diagnosis per the International Classification of Diseases 10th Revision (ICD-10) diagnostic codes coupled with prescription for psychopharmaceuticals per the Anatomical Therapeutic Chemical (ATC) classification codes. **S29 Fig.** Cumulative probability with 95% confidence interval of death with COVID-19 up to 28 days in people with psychotic disorders in epoch 3, cases ascertained by diagnosis per the International Classification of Diseases 10th Revision (ICD-10) diagnostic codes coupled with prescription for psychopharmaceuticals per the Anatomical Therapeutic Chemical (ATC) classification codes. **S30 Fig.** Cumulative probability with 95% confidence interval of death with COVID-19 up to 28 days in people with psychotic disorders in epoch 4, cases ascertained by diagnosis per the International Classification of Diseases 10th Revision (ICD-10) diagnostic codes coupled with prescription for psychopharmaceuticals per the Anatomical Therapeutic Chemical (ATC) classification codes. **S31 Fig.** Cumulative probability with 95% confidence

interval of death with COVID-19 up to 28 days in people with psychotic disorders in epoch 5, cases ascertained by diagnosis per the International Classification of Diseases 10th Revision (ICD-10) diagnostic codes coupled with prescription for psychopharmaceuticals per the Anatomical Therapeutic Chemical (ATC) classification codes. **S32 Fig.** Cumulative probability with 95% confidence interval of death with COVID-19 up to 28 days in people with affective disorders in epoch 1, cases ascertained by diagnosis per the International Classification of Diseases 10th Revision (ICD-10) diagnostic codes. **S33 Fig.** Cumulative probability with 95% confidence interval of death with COVID-19 up to 28 days in people with affective disorders in epoch 2, cases ascertained by diagnosis per the International Classification of Diseases 10th Revision (ICD-10) diagnostic codes. **S34 Fig.** Cumulative probability with 95% confidence interval of death with COVID-19 up to 28 days in people with affective disorders in epoch 3, cases ascertained by diagnosis per the International Classification of Diseases 10th Revision (ICD-10) diagnostic codes. **S35 Fig.** Cumulative probability with 95% confidence interval of death with COVID-19 up to 28 days in people with affective disorders in epoch 4, cases ascertained by diagnosis per the International Classification of Diseases 10th Revision (ICD-10) diagnostic codes. **S36 Fig.** Cumulative probability with 95% confidence interval of death with COVID-19 up to 28 days in people with affective disorders in epoch 5, cases ascertained by diagnosis per the International Classification of Diseases 10th Revision (ICD-10) diagnostic codes. **S37 Fig.** Cumulative probability with 95% confidence interval of death with COVID-19 up to 28 days in people with affective disorders in epoch 1, cases ascertained by diagnosis per the International Classification of Diseases 10th Revision (ICD-10) diagnostic codes coupled with prescription for psychopharmaceuticals per the Anatomical Therapeutic Chemical (ATC) classification codes. **S38 Fig.** Cumulative probability with 95% confidence interval of death with COVID-19 up to 28 days in people with affective disorders in epoch 2, cases ascertained by diagnosis per the International Classification of Diseases 10th Revision (ICD-10) diagnostic codes coupled with prescription for psychopharmaceuticals per the Anatomical Therapeutic Chemical (ATC) classification codes. **S39 Fig.** Cumulative probability with 95% confidence interval of death with COVID-19 up to 28 days in people with affective disorders in epoch 3, cases ascertained by diagnosis per the International Classification of Diseases 10th Revision (ICD-10) diagnostic codes coupled with prescription for psychopharmaceuticals per the Anatomical Therapeutic Chemical (ATC) classification codes. **S40 Fig.** Cumulative probability with 95% confidence interval of death with COVID-19 up to 28 days in people with affective disorders in epoch 4, cases ascertained by diagnosis per the International Classification of Diseases 10th Revision (ICD-10) diagnostic codes coupled with prescription for psychopharmaceuticals per the Anatomical Therapeutic Chemical (ATC) classification codes. **S41 Fig.** Cumulative probability with 95% confidence interval of death with COVID-19 up to 28 days in people with affective disorders in epoch 5, cases ascertained by diagnosis per the International Classification of Diseases 10th Revision (ICD-10) diagnostic codes coupled with prescription for psychopharmaceuticals per the Anatomical Therapeutic Chemical (ATC) classification codes. **S42 Fig.** Cumulative probability with 95% confidence interval of death with COVID-19 up to 28 days in people with anxiety disorders in epoch 1, cases ascertained by diagnosis per the International Classification of Diseases 10th Revision (ICD-10) diagnostic codes. **S43 Fig.** Cumulative probability with 95% confidence interval of death with COVID-19 up to 28 days in people with anxiety disorders in epoch 2, cases ascertained by diagnosis per the International Classification of Diseases 10th Revision (ICD-10) diagnostic codes. **S44 Fig.** Cumulative probability with 95% confidence interval of death with COVID-19 up to 28 days in people with anxiety disorders in epoch 3, cases ascertained by diagnosis per the International Classification of Diseases 10th Revision (ICD-10) diagnostic codes. **S45 Fig.** Cumulative probability with 95% confidence interval of death with COVID-19 up to 28 days in people

with anxiety disorders in epoch 4, cases ascertained by diagnosis per the International Classification of Diseases 10th Revision (ICD-10) diagnostic codes. **S46 Fig.** Cumulative probability with 95% confidence interval of death with COVID-19 up to 28 days in people with anxiety disorders in epoch 5, cases ascertained by diagnosis per the International Classification of Diseases 10th Revision (ICD-10) diagnostic codes. **S47 Fig.** Cumulative probability with 95% confidence interval of death with COVID-19 up to 28 days in people with anxiety disorders in epoch 1, cases ascertained by diagnosis per the International Classification of Diseases 10th Revision (ICD-10) diagnostic codes coupled with prescription for psychopharmaceuticals per the Anatomical Therapeutic Chemical (ATC) classification codes. **S48 Fig.** Cumulative probability with 95% confidence interval of death with COVID-19 up to 28 days in people with anxiety disorders in epoch 2, cases ascertained by diagnosis per the International Classification of Diseases 10th Revision (ICD-10) diagnostic codes coupled with prescription for psychopharmaceuticals per the Anatomical Therapeutic Chemical (ATC) classification codes. **S49 Fig.** Cumulative probability with 95% confidence interval of death with COVID-19 up to 28 days in people with anxiety disorders in epoch 3, cases ascertained by diagnosis per the International Classification of Diseases 10th Revision (ICD-10) diagnostic codes coupled with prescription for psychopharmaceuticals per the Anatomical Therapeutic Chemical (ATC) classification codes. **S50 Fig.** Cumulative probability with 95% confidence interval of death with COVID-19 up to 28 days in people with anxiety disorders in epoch 4, cases ascertained by diagnosis per the International Classification of Diseases 10th Revision (ICD-10) diagnostic codes coupled with prescription for psychopharmaceuticals per the Anatomical Therapeutic Chemical (ATC) classification codes. **S51 Fig.** Cumulative probability with 95% confidence interval of death with COVID-19 up to 28 days in people with anxiety disorders in epoch 5, cases ascertained by diagnosis per the International Classification of Diseases 10th Revision (ICD-10) diagnostic codes coupled with prescription for psychopharmaceuticals per the Anatomical Therapeutic Chemical (ATC) classification codes. **S52 Fig.** Cumulative probability with 95% confidence interval of death with COVID-19 up to 60 days in people with any mental disorder in epoch 1, cases ascertained by diagnosis per the International Classification of Diseases 10th Revision (ICD-10) diagnostic codes. **S53 Fig.** Cumulative probability with 95% confidence interval of death with COVID-19 up to 60 days in people with any mental disorder in epoch 2, cases ascertained by diagnosis per the International Classification of Diseases 10th Revision (ICD-10) diagnostic codes. **S54 Fig.** Cumulative probability with 95% confidence interval of death with COVID-19 up to 60 days in people with any mental disorder in epoch 3, cases ascertained by diagnosis per the International Classification of Diseases 10th Revision (ICD-10) diagnostic codes. **S55 Fig.** Cumulative probability with 95% confidence interval of death with COVID-19 up to 60 days in people with any mental disorder in epoch 4, cases ascertained by diagnosis per the International Classification of Diseases 10th Revision (ICD-10) diagnostic codes. **S56 Fig.** Cumulative probability with 95% confidence interval of death with COVID-19 up to 60 days in people with any mental disorder in epoch 5, cases ascertained by diagnosis per the International Classification of Diseases 10th Revision (ICD-10) diagnostic codes. **S57 Fig.** Cumulative probability with 95% confidence interval of death with COVID-19 up to 60 days in people with any mental disorder in epoch 1, cases ascertained by diagnosis per the International Classification of Diseases 10th Revision (ICD-10) diagnostic codes coupled with prescription for psychopharmaceuticals per the Anatomical Therapeutic Chemical (ATC) classification codes. **S58 Fig.** Cumulative probability with 95% confidence interval of death with COVID-19 up to 60 days in people with any mental disorder in epoch 2, cases ascertained by diagnosis per the International Classification of Diseases 10th Revision (ICD-10) diagnostic codes coupled with prescription for psychopharmaceuticals per the Anatomical Therapeutic Chemical (ATC) classification codes. **S59 Fig.** Cumulative probability with 95% confidence

interval of death with COVID-19 up to 60 days in people with any mental disorder in epoch 3, cases ascertained by diagnosis per the International Classification of Diseases 10th Revision (ICD-10) diagnostic codes coupled with prescription for psychopharmaceuticals per the Anatomical Therapeutic Chemical (ATC) classification codes. **S60 Fig.** Cumulative probability with 95% confidence interval of death with COVID-19 up to 60 days in people with any mental disorder in epoch 4, cases ascertained by diagnosis per the International Classification of Diseases 10th Revision (ICD-10) diagnostic codes coupled with prescription for psychopharmaceuticals per the Anatomical Therapeutic Chemical (ATC) classification codes. **S61 Fig.** Cumulative probability with 95% confidence interval of death with COVID-19 up to 60 days in people with any mental disorder in epoch 5, cases ascertained by diagnosis per the International Classification of Diseases 10th Revision (ICD-10) diagnostic codes coupled with prescription for psychopharmaceuticals per the Anatomical Therapeutic Chemical (ATC) classification codes. **S62 Fig.** Cumulative probability with 95% confidence interval of death with COVID-19 up to 60 days in people with substance use disorders in epoch 1, cases ascertained by diagnosis per the International Classification of Diseases 10th Revision (ICD-10) diagnostic codes. **S63 Fig.** Cumulative probability with 95% confidence interval of death with COVID-19 up to 60 days in people with substance use disorders in epoch 2, cases ascertained by diagnosis per the International Classification of Diseases 10th Revision (ICD-10) diagnostic codes. **S64 Fig.** Cumulative probability with 95% confidence interval of death with COVID-19 up to 60 days in people with substance use disorders in epoch 3, cases ascertained by diagnosis per the International Classification of Diseases 10th Revision (ICD-10) diagnostic codes. **S65 Fig.** Cumulative probability with 95% confidence interval of death with COVID-19 up to 60 days in people with substance use disorders in epoch 4, cases ascertained by diagnosis per the International Classification of Diseases 10th Revision (ICD-10) diagnostic codes. **S66 Fig.** Cumulative probability with 95% confidence interval of death with COVID-19 up to 60 days in people with substance use disorders in epoch 5, cases ascertained by diagnosis per the International Classification of Diseases 10th Revision (ICD-10) diagnostic codes. **S67 Fig.** Cumulative probability with 95% confidence interval of death with COVID-19 up to 60 days in people with substance use disorders in epoch 1, cases ascertained by diagnosis per the International Classification of Diseases 10th Revision (ICD-10) diagnostic codes coupled with prescription for psychopharmaceuticals per the Anatomical Therapeutic Chemical (ATC) classification codes. **S68 Fig.** Cumulative probability with 95% confidence interval of death with COVID-19 up to 60 days in people with substance use disorders in epoch 2, cases ascertained by diagnosis per the International Classification of Diseases 10th Revision (ICD-10) diagnostic codes coupled with prescription for psychopharmaceuticals per the Anatomical Therapeutic Chemical (ATC) classification codes. **S69 Fig.** Cumulative probability with 95% confidence interval of death with COVID-19 up to 60 days in people with substance use disorders in epoch 3, cases ascertained by diagnosis per the International Classification of Diseases 10th Revision (ICD-10) diagnostic codes coupled with prescription for psychopharmaceuticals per the Anatomical Therapeutic Chemical (ATC) classification codes. **S70 Fig.** Cumulative probability with 95% confidence interval of death with COVID-19 up to 60 days in people with substance use disorders in epoch 4, cases ascertained by diagnosis per the International Classification of Diseases 10th Revision (ICD-10) diagnostic codes coupled with prescription for psychopharmaceuticals per the Anatomical Therapeutic Chemical (ATC) classification codes. **S71 Fig.** Cumulative probability with 95% confidence interval of death with COVID-19 up to 60 days in people with substance use disorders in epoch 5, cases ascertained by diagnosis per the International Classification of Diseases 10th Revision (ICD-10) diagnostic codes coupled with prescription for psychopharmaceuticals per the Anatomical Therapeutic Chemical (ATC) classification codes. **S72 Fig.** Cumulative probability with 95% confidence interval of death with COVID-19

up to 60 days in people with psychotic disorders in epoch 1, cases ascertained by diagnosis per the International Classification of Diseases 10th Revision (ICD-10) diagnostic codes. **S73 Fig.** Cumulative probability with 95% confidence interval of death with COVID-19 up to 60 days in people with psychotic disorders in epoch 2, cases ascertained by diagnosis per the International Classification of Diseases 10th Revision (ICD-10) diagnostic codes. **S74 Fig.** Cumulative probability with 95% confidence interval of death with COVID-19 up to 60 days in people with psychotic disorders in epoch 3, cases ascertained by diagnosis per the International Classification of Diseases 10th Revision (ICD-10) diagnostic codes. **S75 Fig.** Cumulative probability with 95% confidence interval of death with COVID-19 up to 60 days in people with psychotic disorders in epoch 4, cases ascertained by diagnosis per the International Classification of Diseases 10th Revision (ICD-10) diagnostic codes. **S76 Fig.** Cumulative probability with 95% confidence interval of death with COVID-19 up to 60 days in people with psychotic disorders in epoch 5, cases ascertained by diagnosis per the International Classification of Diseases 10th Revision (ICD-10) diagnostic codes. **S77 Fig.** Cumulative probability with 95% confidence interval of death with COVID-19 up to 60 days in people with psychotic disorders in epoch 1, cases ascertained by diagnosis per the International Classification of Diseases 10th Revision (ICD-10) diagnostic codes coupled with prescription for psychopharmaceuticals per the Anatomical Therapeutic Chemical (ATC) classification codes. **S78 Fig.** Cumulative probability with 95% confidence interval of death with COVID-19 up to 60 days in people with psychotic disorders in epoch 2, cases ascertained by diagnosis per the International Classification of Diseases 10th Revision (ICD-10) diagnostic codes coupled with prescription for psychopharmaceuticals per the Anatomical Therapeutic Chemical (ATC) classification codes. **S79 Fig.** Cumulative probability with 95% confidence interval of death with COVID-19 up to 60 days in people with psychotic disorders in epoch 3, cases ascertained by diagnosis per the International Classification of Diseases 10th Revision (ICD-10) diagnostic codes coupled with prescription for psychopharmaceuticals per the Anatomical Therapeutic Chemical (ATC) classification codes. **S80 Fig.** Cumulative probability with 95% confidence interval of death with COVID-19 up to 60 days in people with psychotic disorders in epoch 4, cases ascertained by diagnosis per the International Classification of Diseases 10th Revision (ICD-10) diagnostic codes coupled with prescription for psychopharmaceuticals per the Anatomical Therapeutic Chemical (ATC) classification codes. **S81 Fig.** Cumulative probability with 95% confidence interval of death with COVID-19 up to 60 days in people with psychotic disorders in epoch 5, cases ascertained by diagnosis per the International Classification of Diseases 10th Revision (ICD-10) diagnostic codes coupled with prescription for psychopharmaceuticals per the Anatomical Therapeutic Chemical (ATC) classification codes. **S82 Fig.** Cumulative probability with 95% confidence interval of death with COVID-19 up to 60 days in people with affective disorders in epoch 1, cases ascertained by diagnosis per the International Classification of Diseases 10th Revision (ICD-10) diagnostic codes. **S83 Fig.** Cumulative probability with 95% confidence interval of death with COVID-19 up to 60 days in people with affective disorders in epoch 2, cases ascertained by diagnosis per the International Classification of Diseases 10th Revision (ICD-10) diagnostic codes. **S84 Fig.** Cumulative probability with 95% confidence interval of death with COVID-19 up to 60 days in people with affective disorders in epoch 3, cases ascertained by diagnosis per the International Classification of Diseases 10th Revision (ICD-10) diagnostic codes. **S85 Fig.** Cumulative probability with 95% confidence interval of death with COVID-19 up to 60 days in people with affective disorders in epoch 4, cases ascertained by diagnosis per the International Classification of Diseases 10th Revision (ICD-10) diagnostic codes. **S86 Fig.** Cumulative probability with 95% confidence interval of death with COVID-19 up to 60 days in people with affective disorders in epoch 5, cases ascertained by diagnosis per the International Classification of Diseases 10th Revision (ICD-10) diagnostic

codes. **S87 Fig.** Cumulative probability with 95% confidence interval of death with COVID-19 up to 60 days in people with affective disorders in epoch 1, cases ascertained by diagnosis per the International Classification of Diseases 10th Revision (ICD-10) diagnostic codes coupled with prescription for psychopharmaceuticals per the Anatomical Therapeutic Chemical (ATC) classification codes. **S88 Fig.** Cumulative probability with 95% confidence interval of death with COVID-19 up to 60 days in people with affective disorders in epoch 2, cases ascertained by diagnosis per the International Classification of Diseases 10th Revision (ICD-10) diagnostic codes coupled with prescription for psychopharmaceuticals per the Anatomical Therapeutic Chemical (ATC) classification codes. **S89 Fig.** Cumulative probability with 95% confidence interval of death with COVID-19 up to 60 days in people with affective disorders in epoch 3, cases ascertained by diagnosis per the International Classification of Diseases 10th Revision (ICD-10) diagnostic codes coupled with prescription for psychopharmaceuticals per the Anatomical Therapeutic Chemical (ATC) classification codes. **S90 Fig.** Cumulative probability with 95% confidence interval of death with COVID-19 up to 60 days in people with affective disorders in epoch 4, cases ascertained by diagnosis per the International Classification of Diseases 10th Revision (ICD-10) diagnostic codes coupled with prescription for psychopharmaceuticals per the Anatomical Therapeutic Chemical (ATC) classification codes. **S91 Fig.** Cumulative probability with 95% confidence interval of death with COVID-19 up to 60 days in people with affective disorders in epoch 5, cases ascertained by diagnosis per the International Classification of Diseases 10th Revision (ICD-10) diagnostic codes coupled with prescription for psychopharmaceuticals per the Anatomical Therapeutic Chemical (ATC) classification codes. **S92 Fig.** Cumulative probability with 95% confidence interval of death with COVID-19 up to 60 days in people with anxiety disorders in epoch 1, cases ascertained by diagnosis per the International Classification of Diseases 10th Revision (ICD-10) diagnostic codes. **S93 Fig.** Cumulative probability with 95% confidence interval of death with COVID-19 up to 60 days in people with anxiety disorders in epoch 2, cases ascertained by diagnosis per the International Classification of Diseases 10th Revision (ICD-10) diagnostic codes. **S94 Fig.** Cumulative probability with 95% confidence interval of death with COVID-19 up to 60 days in people with anxiety disorders in epoch 3, cases ascertained by diagnosis per the International Classification of Diseases 10th Revision (ICD-10) diagnostic codes. **S95 Fig.** Cumulative probability with 95% confidence interval of death with COVID-19 up to 60 days in people with anxiety disorders in epoch 4, cases ascertained by diagnosis per the International Classification of Diseases 10th Revision (ICD-10) diagnostic codes. **S96 Fig.** Cumulative probability with 95% confidence interval of death with COVID-19 up to 60 days in people with anxiety disorders in epoch 5, cases ascertained by diagnosis per the International Classification of Diseases 10th Revision (ICD-10) diagnostic codes. **S97 Fig.** Cumulative probability with 95% confidence interval of death with COVID-19 up to 60 days in people with anxiety disorders in epoch 1, cases ascertained by diagnosis per the International Classification of Diseases 10th Revision (ICD-10) diagnostic codes coupled with prescription for psychopharmaceuticals per the Anatomical Therapeutic Chemical (ATC) classification codes. **S98 Fig.** Cumulative probability with 95% confidence interval of death with COVID-19 up to 60 days in people with anxiety disorders in epoch 2, cases ascertained by diagnosis per the International Classification of Diseases 10th Revision (ICD-10) diagnostic codes coupled with prescription for psychopharmaceuticals per the Anatomical Therapeutic Chemical (ATC) classification codes. **S99 Fig.** Cumulative probability with 95% confidence interval of death with COVID-19 up to 60 days in people with anxiety disorders in epoch 3, cases ascertained by diagnosis per the International Classification of Diseases 10th Revision (ICD-10) diagnostic codes coupled with prescription for psychopharmaceuticals per the Anatomical Therapeutic Chemical (ATC) classification codes. **S100 Fig.** Cumulative probability with 95% confidence interval of death

with COVID-19 up to 60 days in people with anxiety disorders in epoch 4, cases ascertained by diagnosis per the International Classification of Diseases 10th Revision (ICD-10) diagnostic codes coupled with prescription for psychopharmaceuticals per the Anatomical Therapeutic Chemical (ATC) classification codes. **S101 Fig.** Cumulative probability with 95% confidence interval of death with COVID-19 up to 60 days in people with anxiety disorders in epoch 5, cases ascertained by diagnosis per the International Classification of Diseases 10th Revision (ICD-10) diagnostic codes coupled with prescription for psychopharmaceuticals per the Anatomical Therapeutic Chemical (ATC) classification codes. **S102 Fig.** Cumulative probability with 95% confidence interval of all-cause mortality up to 28 days in people with any mental disorder in epoch 1, cases ascertained by diagnosis per the International Classification of Diseases 10th Revision (ICD-10) diagnostic codes. **S103 Fig.** Cumulative probability with 95% confidence interval of all-cause mortality up to 28 days in people with any mental disorder in epoch 2, cases ascertained by diagnosis per the International Classification of Diseases 10th Revision (ICD-10) diagnostic codes. **S104 Fig.** Cumulative probability with 95% confidence interval of all-cause mortality up to 28 days in people with any mental disorder in epoch 3, cases ascertained by diagnosis per the International Classification of Diseases 10th Revision (ICD-10) diagnostic codes. **S105 Fig.** Cumulative probability with 95% confidence interval of all-cause mortality up to 28 days in people with any mental disorder in epoch 4, cases ascertained by diagnosis per the International Classification of Diseases 10th Revision (ICD-10) diagnostic codes. **S106 Fig.** Cumulative probability with 95% confidence interval of all-cause mortality up to 28 days in people with any mental disorder in epoch 5, cases ascertained by diagnosis per the International Classification of Diseases 10th Revision (ICD-10) diagnostic codes. **S107 Fig.** Cumulative probability with 95% confidence interval of all-cause mortality up to 28 days in people with any mental disorder in epoch 1, cases ascertained by diagnosis per the International Classification of Diseases 10th Revision (ICD-10) diagnostic codes coupled with prescription for psychopharmaceuticals per the Anatomical Therapeutic Chemical (ATC) classification codes. **S108 Fig.** Cumulative probability with 95% confidence interval of all-cause mortality up to 28 days in people with any mental disorder in epoch 2, cases ascertained by diagnosis per the International Classification of Diseases 10th Revision (ICD-10) diagnostic codes coupled with prescription for psychopharmaceuticals per the Anatomical Therapeutic Chemical (ATC) classification codes. **S109 Fig.** Cumulative probability with 95% confidence interval of all-cause mortality up to 28 days in people with any mental disorder in epoch 3, cases ascertained by diagnosis per the International Classification of Diseases 10th Revision (ICD-10) diagnostic codes coupled with prescription for psychopharmaceuticals per the Anatomical Therapeutic Chemical (ATC) classification codes. **S110 Fig.** Cumulative probability with 95% confidence interval of all-cause mortality up to 28 days in people with any mental disorder in epoch 4, cases ascertained by diagnosis per the International Classification of Diseases 10th Revision (ICD-10) diagnostic codes coupled with prescription for psychopharmaceuticals per the Anatomical Therapeutic Chemical (ATC) classification codes. **S111 Fig.** Cumulative probability with 95% confidence interval of all-cause mortality up to 28 days in people with any mental disorder in epoch 5, cases ascertained by diagnosis per the International Classification of Diseases 10th Revision (ICD-10) diagnostic codes coupled with prescription for psychopharmaceuticals per the Anatomical Therapeutic Chemical (ATC) classification codes. **S112 Fig.** Cumulative probability with 95% confidence interval of all-cause mortality up to 28 days in people with substance use disorders in epoch 1, cases ascertained by diagnosis per the International Classification of Diseases 10th Revision (ICD-10) diagnostic codes. **S113 Fig.** Cumulative probability with 95% confidence interval of all-cause mortality up to 28 days in people with substance use disorders in epoch 2, cases ascertained by diagnosis per the International Classification of Diseases 10th Revision (ICD-10) diagnostic

codes. **S114 Fig.** Cumulative probability with 95% confidence interval of all-cause mortality up to 28 days in people with substance use disorders in epoch 3, cases ascertained by diagnosis per the International Classification of Diseases 10th Revision (ICD-10) diagnostic codes. **S115 Fig.** Cumulative probability with 95% confidence interval of all-cause mortality up to 28 days in people with substance use disorders in epoch 4, cases ascertained by diagnosis per the International Classification of Diseases 10th Revision (ICD-10) diagnostic codes. **S116 Fig.** Cumulative probability with 95% confidence interval of all-cause mortality up to 28 days in people with substance use disorders in epoch 5, cases ascertained by diagnosis per the International Classification of Diseases 10th Revision (ICD-10) diagnostic codes. **S117 Fig.** Cumulative probability with 95% confidence interval of all-cause mortality up to 28 days in people with substance use disorders in epoch 1, cases ascertained by diagnosis per the International Classification of Diseases 10th Revision (ICD-10) diagnostic codes coupled with prescription for psychopharmaceuticals per the Anatomical Therapeutic Chemical (ATC) classification codes. **S118 Fig.** Cumulative probability with 95% confidence interval of all-cause mortality up to 28 days in people with substance use disorders in epoch 2, cases ascertained by diagnosis per the International Classification of Diseases 10th Revision (ICD-10) diagnostic codes coupled with prescription for psychopharmaceuticals per the Anatomical Therapeutic Chemical (ATC) classification codes. **S119 Fig.** Cumulative probability with 95% confidence interval of all-cause mortality up to 28 days in people with substance use disorders in epoch 3, cases ascertained by diagnosis per the International Classification of Diseases 10th Revision (ICD-10) diagnostic codes coupled with prescription for psychopharmaceuticals per the Anatomical Therapeutic Chemical (ATC) classification codes. **S120 Fig.** Cumulative probability with 95% confidence interval of all-cause mortality up to 28 days in people with substance use disorders in epoch 4, cases ascertained by diagnosis per the International Classification of Diseases 10th Revision (ICD-10) diagnostic codes coupled with prescription for psychopharmaceuticals per the Anatomical Therapeutic Chemical (ATC) classification codes. **S121 Fig.** Cumulative probability with 95% confidence interval of all-cause mortality up to 28 days in people with substance use disorders in epoch 5, cases ascertained by diagnosis per the International Classification of Diseases 10th Revision (ICD-10) diagnostic codes coupled with prescription for psychopharmaceuticals per the Anatomical Therapeutic Chemical (ATC) classification codes. **S122 Fig.** Cumulative probability with 95% confidence interval of all-cause mortality up to 28 days in people with psychotic disorders in epoch 1, cases ascertained by diagnosis per the International Classification of Diseases 10th Revision (ICD-10) diagnostic codes. **S123 Fig.** Cumulative probability with 95% confidence interval of all-cause mortality up to 28 days in people with psychotic disorders in epoch 2, cases ascertained by diagnosis per the International Classification of Diseases 10th Revision (ICD-10) diagnostic codes. **S124 Fig.** Cumulative probability with 95% confidence interval of all-cause mortality up to 28 days in people with psychotic disorders in epoch 3, cases ascertained by diagnosis per the International Classification of Diseases 10th Revision (ICD-10) diagnostic codes. **S125 Fig.** Cumulative probability with 95% confidence interval of all-cause mortality up to 28 days in people with psychotic disorders in epoch 4, cases ascertained by diagnosis per the International Classification of Diseases 10th Revision (ICD-10) diagnostic codes. **S126 Fig.** Cumulative probability with 95% confidence interval of all-cause mortality up to 28 days in people with psychotic disorders in epoch 5, cases ascertained by diagnosis per the International Classification of Diseases 10th Revision (ICD-10) diagnostic codes. **S127 Fig.** Cumulative probability with 95% confidence interval of all-cause mortality up to 28 days in people with psychotic disorders in epoch 1, cases ascertained by diagnosis per the International Classification of Diseases 10th Revision (ICD-10) diagnostic codes coupled with prescription for psychopharmaceuticals per the Anatomical Therapeutic Chemical (ATC) classification codes. **S128 Fig.** Cumulative probability with 95%

confidence interval of all-cause mortality up to 28 days in people with psychotic disorders in epoch 2, cases ascertained by diagnosis per the International Classification of Diseases 10th Revision (ICD-10) diagnostic codes coupled with prescription for psychopharmaceuticals per the Anatomical Therapeutic Chemical (ATC) classification codes. **S129 Fig.** Cumulative probability with 95% confidence interval of all-cause mortality up to 28 days in people with psychotic disorders in epoch 3, cases ascertained by diagnosis per the International Classification of Diseases 10th Revision (ICD-10) diagnostic codes coupled with prescription for psychopharmaceuticals per the Anatomical Therapeutic Chemical (ATC) classification codes. **S130 Fig.** Cumulative probability with 95% confidence interval of all-cause mortality up to 28 days in people with psychotic disorders in epoch 4, cases ascertained by diagnosis per the International Classification of Diseases 10th Revision (ICD-10) diagnostic codes coupled with prescription for psychopharmaceuticals per the Anatomical Therapeutic Chemical (ATC) classification codes. **S131 Fig.** Cumulative probability with 95% confidence interval of all-cause mortality up to 28 days in people with psychotic disorders in epoch 5, cases ascertained by diagnosis per the International Classification of Diseases 10th Revision (ICD-10) diagnostic codes coupled with prescription for psychopharmaceuticals per the Anatomical Therapeutic Chemical (ATC) classification codes. **S132 Fig.** Cumulative probability with 95% confidence interval of all-cause mortality up to 28 days in people with affective disorders in epoch 1, cases ascertained by diagnosis per the International Classification of Diseases 10th Revision (ICD-10) diagnostic codes. **S133 Fig.** Cumulative probability with 95% confidence interval of all-cause mortality up to 28 days in people with affective disorders in epoch 2, cases ascertained by diagnosis per the International Classification of Diseases 10th Revision (ICD-10) diagnostic codes. **S134 Fig.** Cumulative probability with 95% confidence interval of all-cause mortality up to 28 days in people with affective disorders in epoch 3, cases ascertained by diagnosis per the International Classification of Diseases 10th Revision (ICD-10) diagnostic codes. **S135 Fig.** Cumulative probability with 95% confidence interval of all-cause mortality up to 28 days in people with affective disorders in epoch 4, cases ascertained by diagnosis per the International Classification of Diseases 10th Revision (ICD-10) diagnostic codes. **S136 Fig.** Cumulative probability with 95% confidence interval of all-cause mortality up to 28 days in people with affective disorders in epoch 5, cases ascertained by diagnosis per the International Classification of Diseases 10th Revision (ICD-10) diagnostic codes. **S137 Fig.** Cumulative probability with 95% confidence interval of all-cause mortality up to 28 days in people with affective disorders in epoch 1, cases ascertained by diagnosis per the International Classification of Diseases 10th Revision (ICD-10) diagnostic codes coupled with prescription for psychopharmaceuticals per the Anatomical Therapeutic Chemical (ATC) classification codes. **S138 Fig.** Cumulative probability with 95% confidence interval of all-cause mortality up to 28 days in people with affective disorders in epoch 2, cases ascertained by diagnosis per the International Classification of Diseases 10th Revision (ICD-10) diagnostic codes coupled with prescription for psychopharmaceuticals per the Anatomical Therapeutic Chemical (ATC) classification codes. **S139 Fig.** Cumulative probability with 95% confidence interval of all-cause mortality up to 28 days in people with affective disorders in epoch 3, cases ascertained by diagnosis per the International Classification of Diseases 10th Revision (ICD-10) diagnostic codes coupled with prescription for psychopharmaceuticals per the Anatomical Therapeutic Chemical (ATC) classification codes. **S140 Fig.** Cumulative probability with 95% confidence interval of all-cause mortality up to 28 days in people with affective disorders in epoch 4, cases ascertained by diagnosis per the International Classification of Diseases 10th Revision (ICD-10) diagnostic codes coupled with prescription for psychopharmaceuticals per the Anatomical Therapeutic Chemical (ATC) classification codes. **S141 Fig.** Cumulative probability with 95% confidence interval of all-cause mortality up to 28 days in people with affective disorders in epoch 5, cases

ascertained by diagnosis per the International Classification of Diseases 10th Revision (ICD-10) diagnostic codes coupled with prescription for psychopharmaceuticals per the Anatomical Therapeutic Chemical (ATC) classification codes. **S142 Fig.** Cumulative probability with 95% confidence interval of all-cause mortality up to 28 days in people with anxiety disorders in epoch 1, cases ascertained by diagnosis per the International Classification of Diseases 10th Revision (ICD-10) diagnostic codes. **S143 Fig.** Cumulative probability with 95% confidence interval of all-cause mortality up to 28 days in people with anxiety disorders in epoch 2, cases ascertained by diagnosis per the International Classification of Diseases 10th Revision (ICD-10) diagnostic codes. **S144 Fig.** Cumulative probability with 95% confidence interval of all-cause mortality up to 28 days in people with anxiety disorders in epoch 3, cases ascertained by diagnosis per the International Classification of Diseases 10th Revision (ICD-10) diagnostic codes. **S145 Fig.** Cumulative probability with 95% confidence interval of all-cause mortality up to 28 days in people with anxiety disorders in epoch 4, cases ascertained by diagnosis per the International Classification of Diseases 10th Revision (ICD-10) diagnostic codes. **S146 Fig.** Cumulative probability with 95% confidence interval of all-cause mortality up to 28 days in people with anxiety disorders in epoch 5, cases ascertained by diagnosis per the International Classification of Diseases 10th Revision (ICD-10) diagnostic codes. **S147 Fig.** Cumulative probability with 95% confidence interval of all-cause mortality up to 28 days in people with anxiety disorders in epoch 1, cases ascertained by diagnosis per the International Classification of Diseases 10th Revision (ICD-10) diagnostic codes coupled with prescription for psychopharmaceuticals per the Anatomical Therapeutic Chemical (ATC) classification codes. **S148 Fig.** Cumulative probability with 95% confidence interval of all-cause mortality up to 28 days in people with anxiety disorders in epoch 2, cases ascertained by diagnosis per the International Classification of Diseases 10th Revision (ICD-10) diagnostic codes coupled with prescription for psychopharmaceuticals per the Anatomical Therapeutic Chemical (ATC) classification codes. **S149 Fig.** Cumulative probability with 95% confidence interval of all-cause mortality up to 28 days in people with anxiety disorders in epoch 3, cases ascertained by diagnosis per the International Classification of Diseases 10th Revision (ICD-10) diagnostic codes coupled with prescription for psychopharmaceuticals per the Anatomical Therapeutic Chemical (ATC) classification codes. **S150 Fig.** Cumulative probability with 95% confidence interval of all-cause mortality up to 28 days in people with anxiety disorders in epoch 4, cases ascertained by diagnosis per the International Classification of Diseases 10th Revision (ICD-10) diagnostic codes coupled with prescription for psychopharmaceuticals per the Anatomical Therapeutic Chemical (ATC) classification codes. **S151 Fig.** Cumulative probability with 95% confidence interval of all-cause mortality up to 28 days in people with anxiety disorders in epoch 5, cases ascertained by diagnosis per the International Classification of Diseases 10th Revision (ICD-10) diagnostic codes coupled with prescription for psychopharmaceuticals per the Anatomical Therapeutic Chemical (ATC) classification codes. **S152 Fig.** Cumulative probability with 95% confidence interval of all-cause mortality up to 60 days in people with any mental disorder in epoch 1, cases ascertained by diagnosis per the International Classification of Diseases 10th Revision (ICD-10) diagnostic codes. **S153 Fig.** Cumulative probability with 95% confidence interval of all-cause mortality up to 60 days in people with any mental disorder in epoch 2, cases ascertained by diagnosis per the International Classification of Diseases 10th Revision (ICD-10) diagnostic codes. **S154 Fig.** Cumulative probability with 95% confidence interval of all-cause mortality up to 60 days in people with any mental disorder in epoch 3, cases ascertained by diagnosis per the International Classification of Diseases 10th Revision (ICD-10) diagnostic codes. **S155 Fig.** Cumulative probability with 95% confidence interval of all-cause mortality up to 60 days in people with any mental disorder in epoch 4, cases ascertained by diagnosis per the International Classification of Diseases 10th Revision (ICD-10)

diagnostic codes. **S156 Fig.** Cumulative probability with 95% confidence interval of all-cause mortality up to 60 days in people with any mental disorder in epoch 5, cases ascertained by diagnosis per the International Classification of Diseases 10th Revision (ICD-10) diagnostic codes. **S157 Fig.** Cumulative probability with 95% confidence interval of all-cause mortality up to 60 days in people with any mental disorder in epoch 1, cases ascertained by diagnosis per the International Classification of Diseases 10th Revision (ICD-10) diagnostic codes coupled with prescription for psychopharmaceuticals per the Anatomical Therapeutic Chemical (ATC) classification codes. **S158 Fig.** Cumulative probability with 95% confidence interval of all-cause mortality up to 60 days in people with any mental disorder in epoch 2, cases ascertained by diagnosis per the International Classification of Diseases 10th Revision (ICD-10) diagnostic codes coupled with prescription for psychopharmaceuticals per the Anatomical Therapeutic Chemical (ATC) classification codes. **S159 Fig.** Cumulative probability with 95% confidence interval of all-cause mortality up to 60 days in people with any mental disorder in epoch 3, cases ascertained by diagnosis per the International Classification of Diseases 10th Revision (ICD-10) diagnostic codes coupled with prescription for psychopharmaceuticals per the Anatomical Therapeutic Chemical (ATC) classification codes. **S160 Fig.** Cumulative probability with 95% confidence interval of all-cause mortality up to 60 days in people with any mental disorder in epoch 4, cases ascertained by diagnosis per the International Classification of Diseases 10th Revision (ICD-10) diagnostic codes coupled with prescription for psychopharmaceuticals per the Anatomical Therapeutic Chemical (ATC) classification codes. **S161 Fig.** Cumulative probability with 95% confidence interval of all-cause mortality up to 60 days in people with any mental disorder in epoch 5, cases ascertained by diagnosis per the International Classification of Diseases 10th Revision (ICD-10) diagnostic codes coupled with prescription for psychopharmaceuticals per the Anatomical Therapeutic Chemical (ATC) classification codes. **S162 Fig.** Cumulative probability with 95% confidence interval of all-cause mortality up to 60 days in people with substance use disorders in epoch 1, cases ascertained by diagnosis per the International Classification of Diseases 10th Revision (ICD-10) diagnostic codes. **S163 Fig.** Cumulative probability with 95% confidence interval of all-cause mortality up to 60 days in people with substance use disorders in epoch 2, cases ascertained by diagnosis per the International Classification of Diseases 10th Revision (ICD-10) diagnostic codes. **S164 Fig.** Cumulative probability with 95% confidence interval of all-cause mortality up to 60 days in people with substance use disorders in epoch 3, cases ascertained by diagnosis per the International Classification of Diseases 10th Revision (ICD-10) diagnostic codes. **S165 Fig.** Cumulative probability with 95% confidence interval of all-cause mortality up to 60 days in people with substance use disorders in epoch 4, cases ascertained by diagnosis per the International Classification of Diseases 10th Revision (ICD-10) diagnostic codes. **S166 Fig.** Cumulative probability with 95% confidence interval of all-cause mortality up to 60 days in people with substance use disorders in epoch 5, cases ascertained by diagnosis per the International Classification of Diseases 10th Revision (ICD-10) diagnostic codes. **S167 Fig.** Cumulative probability with 95% confidence interval of all-cause mortality up to 60 days in people with substance use disorders in epoch 1, cases ascertained by diagnosis per the International Classification of Diseases 10th Revision (ICD-10) diagnostic codes coupled with prescription for psychopharmaceuticals per the Anatomical Therapeutic Chemical (ATC) classification codes. **S168 Fig.** Cumulative probability with 95% confidence interval of all-cause mortality up to 60 days in people with substance use disorders in epoch 2, cases ascertained by diagnosis per the International Classification of Diseases 10th Revision (ICD-10) diagnostic codes coupled with prescription for psychopharmaceuticals per the Anatomical Therapeutic Chemical (ATC) classification codes. **S169 Fig.** Cumulative probability with 95% confidence interval of all-cause mortality up to 60 days in people with substance use disorders in epoch 3, cases ascertained by

diagnosis per the International Classification of Diseases 10th Revision (ICD-10) diagnostic codes coupled with prescription for psychopharmaceuticals per the Anatomical Therapeutic Chemical (ATC) classification codes. **S170 Fig.** Cumulative probability with 95% confidence interval of all-cause mortality up to 60 days in people with substance use disorders in epoch 4, cases ascertained by diagnosis per the International Classification of Diseases 10th Revision (ICD-10) diagnostic codes coupled with prescription for psychopharmaceuticals per the Anatomical Therapeutic Chemical (ATC) classification codes. **S171 Fig.** Cumulative probability with 95% confidence interval of all-cause mortality up to 60 days in people with substance use disorders in epoch 5, cases ascertained by diagnosis per the International Classification of Diseases 10th Revision (ICD-10) diagnostic codes coupled with prescription for psychopharmaceuticals per the Anatomical Therapeutic Chemical (ATC) classification codes. **S172 Fig.** Cumulative probability with 95% confidence interval of all-cause mortality up to 60 days in people with psychotic disorders in epoch 1, cases ascertained by diagnosis per the International Classification of Diseases 10th Revision (ICD-10) diagnostic codes. **S173 Fig.** Cumulative probability with 95% confidence interval of all-cause mortality up to 60 days in people with psychotic disorders in epoch 2, cases ascertained by diagnosis per the International Classification of Diseases 10th Revision (ICD-10) diagnostic codes. **S174 Fig.** Cumulative probability with 95% confidence interval of all-cause mortality up to 60 days in people with psychotic disorders in epoch 3, cases ascertained by diagnosis per the International Classification of Diseases 10th Revision (ICD-10) diagnostic codes. **S175 Fig.** Cumulative probability with 95% confidence interval of all-cause mortality up to 60 days in people with psychotic disorders in epoch 4, cases ascertained by diagnosis per the International Classification of Diseases 10th Revision (ICD-10) diagnostic codes. **S176 Fig.** Cumulative probability with 95% confidence interval of all-cause mortality up to 60 days in people with psychotic disorders in epoch 5, cases ascertained by diagnosis per the International Classification of Diseases 10th Revision (ICD-10) diagnostic codes. **S177 Fig.** Cumulative probability with 95% confidence interval of all-cause mortality up to 60 days in people with psychotic disorders in epoch 1, cases ascertained by diagnosis per the International Classification of Diseases 10th Revision (ICD-10) diagnostic codes coupled with prescription for psychopharmaceuticals per the Anatomical Therapeutic Chemical (ATC) classification codes. **S178 Fig.** Cumulative probability with 95% confidence interval of all-cause mortality up to 60 days in people with psychotic disorders in epoch 2, cases ascertained by diagnosis per the International Classification of Diseases 10th Revision (ICD-10) diagnostic codes coupled with prescription for psychopharmaceuticals per the Anatomical Therapeutic Chemical (ATC) classification codes. **S179 Fig.** Cumulative probability with 95% confidence interval of all-cause mortality up to 60 days in people with psychotic disorders in epoch 3, cases ascertained by diagnosis per the International Classification of Diseases 10th Revision (ICD-10) diagnostic codes coupled with prescription for psychopharmaceuticals per the Anatomical Therapeutic Chemical (ATC) classification codes. **S180 Fig.** Cumulative probability with 95% confidence interval of all-cause mortality up to 60 days in people with psychotic disorders in epoch 4, cases ascertained by diagnosis per the International Classification of Diseases 10th Revision (ICD-10) diagnostic codes coupled with prescription for psychopharmaceuticals per the Anatomical Therapeutic Chemical (ATC) classification codes. **S181 Fig.** Cumulative probability with 95% confidence interval of all-cause mortality up to 60 days in people with psychotic disorders in epoch 5, cases ascertained by diagnosis per the International Classification of Diseases 10th Revision (ICD-10) diagnostic codes coupled with prescription for psychopharmaceuticals per the Anatomical Therapeutic Chemical (ATC) classification codes. **S182 Fig.** Cumulative probability with 95% confidence interval of all-cause mortality up to 60 days in people with affective disorders in epoch 1, cases ascertained by diagnosis per the International Classification of Diseases 10th Revision (ICD-

10) diagnostic codes. **S183 Fig.** Cumulative probability with 95% confidence interval of all-cause mortality up to 60 days in people with affective disorders in epoch 2, cases ascertained by diagnosis per the International Classification of Diseases 10th Revision (ICD-10) diagnostic codes. **S184 Fig.** Cumulative probability with 95% confidence interval of all-cause mortality up to 60 days in people with affective disorders in epoch 3, cases ascertained by diagnosis per the International Classification of Diseases 10th Revision (ICD-10) diagnostic codes. **S185 Fig.** Cumulative probability with 95% confidence interval of all-cause mortality up to 60 days in people with affective disorders in epoch 4, cases ascertained by diagnosis per the International Classification of Diseases 10th Revision (ICD-10) diagnostic codes. **S186 Fig.** Cumulative probability with 95% confidence interval of all-cause mortality up to 60 days in people with affective disorders in epoch 5, cases ascertained by diagnosis per the International Classification of Diseases 10th Revision (ICD-10) diagnostic codes. **S187 Fig.** Cumulative probability with 95% confidence interval of all-cause mortality up to 60 days in people with affective disorders in epoch 1, cases ascertained by diagnosis per the International Classification of Diseases 10th Revision (ICD-10) diagnostic codes coupled with prescription for psychopharmaceuticals per the Anatomical Therapeutic Chemical (ATC) classification codes. **S188 Fig.** Cumulative probability with 95% confidence interval of all-cause mortality up to 60 days in people with affective disorders in epoch 2, cases ascertained by diagnosis per the International Classification of Diseases 10th Revision (ICD-10) diagnostic codes coupled with prescription for psychopharmaceuticals per the Anatomical Therapeutic Chemical (ATC) classification codes. **S189 Fig.** Cumulative probability with 95% confidence interval of all-cause mortality up to 60 days in people with affective disorders in epoch 3, cases ascertained by diagnosis per the International Classification of Diseases 10th Revision (ICD-10) diagnostic codes coupled with prescription for psychopharmaceuticals per the Anatomical Therapeutic Chemical (ATC) classification codes. **S190 Fig.** Cumulative probability with 95% confidence interval of all-cause mortality up to 60 days in people with affective disorders in epoch 4, cases ascertained by diagnosis per the International Classification of Diseases 10th Revision (ICD-10) diagnostic codes coupled with prescription for psychopharmaceuticals per the Anatomical Therapeutic Chemical (ATC) classification codes. **S191 Fig.** Cumulative probability with 95% confidence interval of all-cause mortality up to 60 days in people with affective disorders in epoch 5, cases ascertained by diagnosis per the International Classification of Diseases 10th Revision (ICD-10) diagnostic codes coupled with prescription for psychopharmaceuticals per the Anatomical Therapeutic Chemical (ATC) classification codes. **S192 Fig.** Cumulative probability with 95% confidence interval of all-cause mortality up to 60 days in people with anxiety disorders in epoch 1, cases ascertained by diagnosis per the International Classification of Diseases 10th Revision (ICD-10) diagnostic codes. **S193 Fig.** Cumulative probability with 95% confidence interval of all-cause mortality up to 60 days in people with anxiety disorders in epoch 2, cases ascertained by diagnosis per the International Classification of Diseases 10th Revision (ICD-10) diagnostic codes. **S194 Fig.** Cumulative probability with 95% confidence interval of all-cause mortality up to 60 days in people with anxiety disorders in epoch 3, cases ascertained by diagnosis per the International Classification of Diseases 10th Revision (ICD-10) diagnostic codes. **S195 Fig.** Cumulative probability with 95% confidence interval of all-cause mortality up to 60 days in people with anxiety disorders in epoch 4, cases ascertained by diagnosis per the International Classification of Diseases 10th Revision (ICD-10) diagnostic codes. **S196 Fig.** Cumulative probability with 95% confidence interval of all-cause mortality up to 60 days in people with anxiety disorders in epoch 5, cases ascertained by diagnosis per the International Classification of Diseases 10th Revision (ICD-10) diagnostic codes. **S197 Fig.** Cumulative probability with 95% confidence interval of all-cause mortality up to 60 days in people with anxiety disorders in epoch 1, cases ascertained by diagnosis per the International Classification

of Diseases 10th Revision (ICD-10) diagnostic codes coupled with prescription for psycho-pharmaceuticals per the Anatomical Therapeutic Chemical (ATC) classification codes. **S198 Fig.** Cumulative probability with 95% confidence interval of all-cause mortality up to 60 days in people with anxiety disorders in epoch 2, cases ascertained by diagnosis per the International Classification of Diseases 10th Revision (ICD-10) diagnostic codes coupled with prescription for psychopharmaceuticals per the Anatomical Therapeutic Chemical (ATC) classification codes. **S199 Fig.** Cumulative probability with 95% confidence interval of all-cause mortality up to 60 days in people with anxiety disorders in epoch 3, cases ascertained by diagnosis per the International Classification of Diseases 10th Revision (ICD-10) diagnostic codes coupled with prescription for psychopharmaceuticals per the Anatomical Therapeutic Chemical (ATC) classification codes. **S200 Fig.** Cumulative probability with 95% confidence interval of all-cause mortality up to 60 days in people with anxiety disorders in epoch 4, cases ascertained by diagnosis per the International Classification of Diseases 10th Revision (ICD-10) diagnostic codes coupled with prescription for psychopharmaceuticals per the Anatomical Therapeutic Chemical (ATC) classification codes. **S201 Fig.** Cumulative probability with 95% confidence interval of all-cause mortality up to 60 days in people with anxiety disorders in epoch 5, cases ascertained by diagnosis per the International Classification of Diseases 10th Revision (ICD-10) diagnostic codes coupled with prescription for psychopharmaceuticals per the Anatomical Therapeutic Chemical (ATC) classification codes.
(ZIP)

## Author Contributions

**Conceptualization:** Tomáš Formánek, Libor Potočár, Katrin Wolfova, Hana Melicharová, Karolína Mladá, Anna Wiedemann, Danni Chen, Pavel Mohr, Petr Winkler, Peter B. Jones, Jiří Jarkovský.

**Data curation:** Hana Melicharová, Jiří Jarkovský.

**Formal analysis:** Tomáš Formánek, Libor Potočár, Hana Melicharová, Jiří Jarkovský.

**Investigation:** Libor Potočár.

**Methodology:** Tomáš Formánek, Libor Potočár, Katrin Wolfova, Hana Melicharová, Karolína Mladá, Anna Wiedemann, Danni Chen, Pavel Mohr, Petr Winkler, Peter B. Jones, Jiří Jarkovský.

**Project administration:** Tomáš Formánek.

**Supervision:** Peter B. Jones.

**Validation:** Tomáš Formánek.

**Visualization:** Libor Potočár.

**Writing – original draft:** Tomáš Formánek.

**Writing – review & editing:** Libor Potočár, Katrin Wolfova, Hana Melicharová, Karolína Mladá, Anna Wiedemann, Danni Chen, Pavel Mohr, Petr Winkler, Peter B. Jones, Jiří Jarkovský.

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
