## [Editor Report · Decision Letter 0]

6 Mar 2024

Dear Dr Formánek, 

Thank you for submitting your manuscript entitled "Mortality from COVID-19 and from All-causes following First-ever SARS‑CoV‑2 Infection in Individuals with Pre-existing Mental Disorders: A National Cohort Study" for consideration by PLOS Medicine.

Your manuscript has now been evaluated by the PLOS Medicine editorial staff and I am writing to let you know that we would like to send your submission out for external peer review.

Please re-submit your manuscript within two working days, i.e. by Mar 08 2024.

Feel free to email me at aschaefer@plos.org or us at plosmedicine@plos.org if you have any queries relating to your submission.

Kind regards,

Alexandra Schaefer, PhD

Associate Editor

PLOS Medicine

---

## [Decision Letter · Decision Letter 1]

27 Mar 2024

Dear Dr. Formánek,

Thank you very much for submitting your manuscript "Mortality from COVID-19 and from All-causes following First-ever SARS‑CoV‑2 Infection in Individuals with Pre-existing Mental Disorders: A National Cohort Study" (PMEDICINE-D-24-00730R1) for consideration at PLOS Medicine. 

Your paper was evaluated by an associate editor and discussed among all the editors here. It was also discussed with an academic editor with relevant expertise, and sent to independent reviewers, including a statistical reviewer. The reviews are appended at the bottom of this email and any accompanying reviewer attachments can be seen via the link below:

[LINK]

In light of these reviews, I am afraid that we will not be able to accept the manuscript for publication in the journal in its current form, but we would like to consider a revised version that addresses the reviewers' and editors' comments. Obviously we cannot make any decision about publication until we have seen the revised manuscript and your response, and we plan to seek re-review by one or more of the reviewers. 

Please use the following link to submit the revised manuscript: https://www.editorialmanager.com/pmedicine/

We expect to receive your revised manuscript by Apr 17 2024. However, if this deadline is not feasible, please contact me by email, and we can discuss a suitable alternative.

Don't hesitate to contact me directly with any questions (aschaefer@plos.org). If you reply directly to this message, please be sure to 'Reply All' so your message comes directly to my inbox.

We look forward to receiving your revised manuscript. 

Sincerely,

Alexandra Schaefer, PhD

PLOS Medicine

plosmedicine.org

***Please note: not all will apply to your paper, but please check each item carefully

EDITORIAL COMMENTS

In line with the reviewer's comments, and given that you do not have the data, we feel that the lack of adjustment for health-related confounders such as body mass index as a limitation needs to be discussed in more detail. We also believe that the presentation of data from the Czech Republic is a strength of your study, and you may wish to include points about this aspect and mental health care in the Central and Eastern European region in the introduction and/or discussion.

GENERAL COMMENTS

Please include page numbers and line numbers in the manuscript file. Use continuous line numbers (do not restart the numbering on each page).

FINANCIAL DISCLOSURE

The funding statement should include: specific grant numbers, initials of authors who received each award, URLs to sponsors’ websites. Also, please state whether any sponsors or funders (other than the named authors) played any role in study design, data collection and analysis, the decision to publish, or preparation of the manuscript. If they had no role in the research, include this sentence: “The funders had no role in study design, data collection and analysis, decision to publish, or preparation of the manuscript.”

COMPETING INTEREST

All authors must declare their relevant competing interests per the PLOS policy, which can be seen here:

https://journals.plos.org/plosmedicine/s/competing-interests

For authors with ties to industry, please indicate whether any of the interests has a financial stake in the results of the current study.

DATA AVAILABILITY STATEMENT

The Data Availability Statement (DAS) requires revision. If the data are not freely available, please describe briefly the ethical, legal, or contractual restriction that prevents you from sharing it. Please also include an appropriate contact (web or email address) for inquiries (this cannot be a study author).

TITLE

We recommend including the country in the title. For example: Mortality from COVID-19 and from All-causes following First-ever SARS‑ CoV‑ 2 Infection in Individuals with Pre-existing Mental Disorders: A National Cohort Study from the Czech Republic

ABSTRACT

1) Please structure your abstract using the PLOS Medicine headings (Background, Methods and Findings, Conclusions).

2) PLOS Medicine requests that main results are quantified with 95% CIs as well as p values. When reporting p values please report as p<0.001 and where higher as the exact p value p=0.002, for example. For the purposes of transparent data reporting, if not including the aforementioned please clearly state the reasons why not. When a p value is given, please specify the statistical test used to determine it.

3) Throughout, suggest reporting statistical information as follows to improve clarity for the reader “22% (95% CI [13%,28%]; p</=)”. Please be sure to define all numerical values at first use. Please amend throughout the abstract and main manuscript. Please note the use of commas to separate upper and lower bounds, as opposed to hyphens as these can be confused with reporting of negative values.

3) Please ensure that all numbers presented in the abstract are present and identical to numbers presented in the main manuscript text.

4) Please include the study design, population and setting, number of participants, years during which the study took place, length of follow up, main outcome measures.

5) Please include the important dependent variables that are adjusted for in the analyses.

6) Please define all abbreviations including those for statistical reporting at first use.

7) In the last sentence of the Abstract Methods and Findings section, please describe the main limitation(s) of the study's methodology.

AUTHOR SUMMARY

At this stage, we ask that you include a short, non-technical Author Summary of your research to make findings accessible to a wide audience that includes both scientists and non-scientists. The Author Summary should immediately follow the Abstract in your revised manuscript. This text is subject to editorial change and should be distinct from the scientific abstract. Ideally each sub-heading should contain 2-3 single sentence, concise bullet points containing the most salient points from your study. In the final bullet point of ‘What Do These Findings Mean?’, please include the main limitations of the study in non-technical language. Please see our author guidelines for more information: https://journals.plos.org/plosmedicine/s/revising-your-manuscript#loc-author-summary

METHODS AND RESULTS

1) Please ensure that the study is reported according to the RECORD (STROBE if you feel it is more appropriate) guideline, and include the completed RECORD checklist as Supporting Information. Please add the following statement, or similar, to the Methods: "This study is reported as per the Reporting of studies Conducted using Observational Routinely-collected health Data (RECORD) Statement (S1 Checklist)."

2) PLOS Medicine requests that main results are quantified with 95% CIs as well as p values. We suggest reporting statistical information as detailed above – see under ABSTRACT

3) Please present numerators and denominators for percentages (at least in the Tables [not necessarily each time they're mentioned]).

4) Did your study have a prospective protocol or analysis plan? Please state this (either way) early in the Methods section.

c) In either case, changes in the analysis-- including those made in response to peer review comments-- should be identified as such in the Methods section of the paper, with rationale."

5) For all observational studies, in the manuscript text, please indicate: (1) the specific hypotheses you intended to test, (2) the analytical methods by which you planned to test them, (3) the analyses you actually performed, and (4) when reported analyses differ from those that were planned, transparent explanations for differences that affect the reliability of the study's results. If a reported analysis was performed based on an interesting but unanticipated pattern in the data, please be clear that the analysis was data-driven.

DISCUSSION

Please present and organize the Discussion as follows: a short, clear summary of the article's findings; what the study adds to existing research and where and why the results may differ from previous research; strengths and limitations of the study; implications and next steps for research, clinical practice, and/or public policy; one-paragraph conclusion (no subheadings).

FIGURES AND TABLES

1) Please provide titles and legends for all figures and tables (including those in Supporting Information files).

2) Please define all abbreviations used in each figure/table (including those in Supporting Information files).

3) Please consider avoiding the use of red and green in order to make your figure more accessible to those with color blindness.

SUPPLEMENTARY MATERIAL

1) We suggest reporting statistical information as detailed above – see under ABSTRACT. Please be sure to define all numerical values.

2) As for the main manuscript, please indicate whether analyses are adjusted to help facilitate transparent data reporting please also detail the factors adjusted for and present the unadjusted analyses for comparison. If not, please clearly state the reasons why not.

3) Please cite your Supporting Information as outlined here: https://journals.plos.org/plosmedicine/s/supporting-information

REFERENCES

1) PLOS uses the numbered citation (citation-sequence) method and first six authors, et al.

2) Please ensure that journal name abbreviations match those found in the National Center for Biotechnology Information (NCBI) databases (http://www.ncbi.nlm.nih.gov/nlmcatalog/journals), and are appropriately formatted and capitalised.

3) Where website addresses are cited, please specify the date of access (e.g. [accessed: 16/09/2023]).

4) Please also see https://journals.plos.org/plosmedicine/s/submission-guidelines#loc-references for further details on reference formatting.

Comments from the reviewers:

Reviewer #1: Thank you for the opportunity to review this manuscript. The authors examined associations between mental disorder diagnoses and mortality following SARS-CoV-2 infection in the full Czech population. The manuscript assesses an important epidemiological relationship and stratified by time across the pandemic; however, there are several places where clarification, further description, or modification is required.

- I would clarify in the abstract that psychiatric disorders were defined two ways, first by ICD codes and second by ICD plus prescriptions - as written it seems it is only the second definition which could be restrictive.

- The abstract should briefly describe the findings for all psychiatric disorder groups assessed - as it current does not present affective disorders.

- I don't necessarily think vaccination status can be considered a true confounder, which should precede the exposure and be associated with the outcome (e.g., sex/gender); while vaccination status is likely related to psychiatric disorders, in this sample it seems that psychiatric diagnoses were believed to precede vaccination (e.g., arguably it is possible vaccination is on the pathway between psychiatric diagnoses and covid outcomes). Similarly for physical comorbidities - depending on the timing, it is possible these preceded or occurred after psychiatric disorder onset. I don't think it is necessarily invalid to adjust for these factors (particularly comorbidities depending on their timing of diagnosis), but the authors should describe more precisely how they conceptualize these variables in the causal structure of relationships and why they are adjustment for them.

- What was the reason behind including individuals aged 10 and above?

- Potential limitation of under ascertainment of COVID-19 cases if individuals did not have data in the nationwide testing - were there at home tests available?

- What is the distinction between ICD-10 cause and external cause of death?

- The ICD codes used for the psychiatric disorders should be listed in the supplement. Were stress-related disorders included (e.g., was PTSD included in anxiety disorders)?

- What was the justification behind including the second definition of psychiatric disorders (i.e., plus prescriptions) - do the authors think this identified a subgroup with more severe disease requiring rx treatment? It is also possible that these individuals have better managed psychiatric symptoms if they are receiving efficacious medications, so somewhat hard to interpret.

- How was vaccination status defined - any vaccine at least 14 days after, or full vaccine regimen? Any distinction by vaccination type (depending on what was available) or booster doses?

- How might having unmatched psychiatric patients impact the findings? Or how do the authors interpret the lack of matched non-psychiatric patients for these individuals?

- More detail should be added in the description of the negative control analyses in the main paper description - it is currently a bit unclear what the actual analyses are. E.g., these seem to be negative control exposures, but it is unclear how there are specific proportions for the studied disorders (i.e., the psychiatric disorders)?

- When discussing the previous findings of antipsychotics and decreased mortality in individuals with psychosis, it would be helpful to explicitly describe the current findings for psychosis plus medications, and potentially sensitivity analyses directly testing this same hypothesis (psychosis plus the same antipsychotic tested in the prior study).

- The results section should describe and discussion should interpret associations with "any mental disorder", since these were assessed.

Reviewer #2: "Mortality from COVID-19 and from All-causes following First-ever SARS‑CoV-2 Infection in Individuals with Pre-existing Mental Disorders: A National Cohort Study" investigates the effect on survival rate on people with pre-existing mental disorders, following SARS‑CoV-2 infection, particularly before the availability of vaccines, on Czech national data. Sensitivity analysis was performed to examine the effect of collider bias. It was concluded that pre-existing mental disorders and substance use disorders was correlated with an elevated risk of death following SARS-CoV-2 infection.

1. In the Methods section, it is stated that exact matching was performed

---

## [Decision Letter · Decision Letter 2]

15 May 2024

Dear Dr. Formánek,

Thank you very much for re-submitting your manuscript "Mortality from COVID-19 and from All-causes following First-ever SARS‑CoV‑2 Infection in Individuals with Pre-existing Mental Disorders: A National Cohort Study from Czechia" (PMEDICINE-D-24-00730R2) for review by PLOS Medicine.

Thank you for your detailed response to the editors' and reviewers' comments. I have discussed the paper with my colleagues and the academic editor, and it has also been seen again by two of the original reviewers. The changes made to the paper were satisfactory to the reviewers. As such, we intend to accept the paper for publication, pending your attention to the editorial comments below in a further revision. When submitting your revised paper, please once again include a detailed point-by-point response to the editorial comments.

[LINK]

In revising the manuscript for further consideration here, please ensure you address the specific points made by each reviewer and the editors. In your rebuttal letter you should indicate your response to the reviewers' and editors' comments and the changes you have made in the manuscript. Please submit a clean version of the paper as the main article file. A version with changes marked must also be uploaded as a marked up manuscript file. Please also check the guidelines for revised papers at http://journals.plos.org/plosmedicine/s/revising-your-manuscript for any that apply to your paper. 

We ask that you submit your revision within 1 week (May 22 2024). However, if this deadline is not feasible, please contact me by email, and we can discuss a suitable alternative.

Please do not hesitate to contact me directly with any questions (aschaefer@plos.org). If you reply directly to this message, please be sure to 'Reply All' so your message comes directly to my inbox.

We look forward to receiving the revised manuscript.

Sincerely,

Alexandra Schaefer, PhD

Associate Editor 

PLOS Medicine

plosmedicine.org

Requests from Editors:

GENERAL COMMENTS

We kindly ask you to reduce the number of supplementary figures and combining multiple graphs into one figure, for example, one figure showing all graphs for "28 days post-infection, epochs 1 through 5, affective disorders, diagnosed and treated". We do not have a formal limit on supplemental display items, but we feel that more than 200 figures is excessive for readers to navigate.

When reporting statistical information, please use commas to separate upper and lower bounds, as opposed to hyphens as these can be confused with reporting of negative values.

ABSTRACT

1) l.55: Please simplify the description of the database. We suggest changing “We used Czech national, whole population, all healthcare encompassing register-based data…” to “We used population-level data from a Czech healthcare registry…” (or similar).

2) l.56: Please include the time frames of the five epochs.

3) l.58: Please define ‘ICD’ at first use.

4) l.61: Please define ‘ATC’ at first use.

5) l.61: Please remove the word ‘exact’ (“We matched…").

6) ll.63-64: Please change to “death with COVID-19” (please edit this consistently throughout the paper).

7) l.66: Please briefly state the number of people included in the study together with their demographics (age, sex etc.).

8) l.67: As above “dying from COVID-19” – Is it 100% certain that these people died from COVID-19 or was it possibly death with COVID-19?

9) ll.68 ff: Please present the values for each of the epochs instead of using a “from..to” format. Please revise throughout the entire Abstract.

10) l.77: “COVID-19 death” – Please change to “death with COVID-19”.

11) l.80: Please change to “death with COVID-19”.

12) ll.80-81: “all-cased death” – please revise.

13) In the last sentence of the Abstract Methods and Findings section, please describe the main limitation(s) of the study's methodology.

AUTHOR SUMMARY

1) ll.92-93: Please change to: “Existing research has demonstrated consistently elevated risk of death with COVID-19 or all-cause mortality in people..”. Please revise throughout the entire Author Summary.

2) ll.112-113: Please change to: “These associations could not be fully explained by differences in vaccination uptake or clinically-recorded physical comorbidities.”

3) In the final bullet point of ‘What Do These Findings Mean?’, please describe the main limitations of the study in non-technical language.

METHODS AND RESULTS

1) ll.185-186: The following sentence is unclear: “The Czech National Vaccination Strategy was launched in December 2020, and it considered no pre-existing mental disorders as reason for priority inoculation [32].” Does this mean it did not consider pre-existing mental disorders as a reason for priority inoculation (as written, it suggests that people without preexisting mental disorders were given priority)? Please revise for clarity. 

2) l.199: “The register covers nearly the entire Czech population.” – Is it possible to provide actual numbers here?

3) l.211: Should this be “IHIS” instead of “NHIS”?

4) l.260: Please change “exactly-matched” to “matched” (also at l.497).

5) ll.262-263: Please change “Since vaccination do not confer…” to “Since vaccination does not confer…”

6) ll.264-266: Please change to “For instance, when an individual received the first dose of a two-dose regimen more than 14 days before the infection, and the second dose 14 or less days before the infection…”

7) l.285 ff: "We considered (1) death due to COVID-19 (ICD-10 codes U071 and U072)." - We feel that it should be made clear to the reader that these codes do not identify the cause of death, but rather that a person was diagnosed with COVID-19 with or without the virus identified. Therefore, we feel that "Death with COVID-19" (or e.g. “death in a person diagnosed with COVID-19”) along with an explanation that other underlying causes may have been the actual cause of the death would be a more appropriate description. While we understand that you are trying to rule out competing causes of death (as a sensitivity analysis), we feel that the description needs to be much clearer. Is there an entry on death certificates that states U071 or U072 in a specific cause of death field?

8) l.285 ff: Given that the ICD-10 code "U072" is used to identify an individual whose COVID-19 disease has not yet been confirmed by laboratory testing, we wonder how you have calculated the time period for individuals without a COVID-19 diagnosis (i.e., when does the time period begin)?

9) l.335: Please define ‘SD’ at first use.

10) l.340: Please change to: “Risk of COVID-19 Related Death in People with Pre-existing Mental Disorders”

11) l.341 ff: Please refrain from writing “COVID-19 death”, but use “COVID-19 related death” or similar. Please revise throughout the main text.

12) l.341 ff: We feel it could be misleading to only present values from a specific epoch. We therefore suggest either presenting all values for the epochs discussed or removing the values and referring to the figure and/or table where readers may find the values in full. For readability, we prefer referring to the relevant figures and/or tables. Please revise throughout the manuscript.

13) l.343: Please clearly describe which epochs were covered for the different disorders. For example, you write "in each of the studied epochs," which may lead readers to think of all five epochs, whereas for psychotic disorders only epochs 2, 3, 4, and 5 were covered in the analysis.

14) l.345 ff: Please ensure to include the relevant statistical details in each pair of parenthesis (e.g. l.345 “..and 60 days (aHR 1.46 [95% CIs, 1.20-1.79] to 1.93 [95% CIs, 1.14-3.27]) following..”). Please revise throughout the main text.

15) l.390 ff: We find the presentation of the sensitivity analysis data difficult to understand. As above, we suggest either presenting the data values in full or referring to a relevant table and/or figure where the data can be seen in full. In addition, we feel that the description of the sensitivity analyses in the supplementary material should be incorporated into the main text (ll.308-320), as it is not significantly longer and provides important details, particularly about the selected negative control characteristics. Please also briefly explain how the theoretical maximum of 40 is calculated.

DISCUSSION

1) Please remove any subheadings.

2) ll.491-493: Please change to “Strengths included..”. We suggest describing the strengths of the study in more detail as currently the description “in a complex and detailed way” is rather vague.

FIGURES AND TABLES

1) Table 1/2: Please ensure to define all abbreviations used in the table below the table (SD, Inf., IQR, ICD, ATC). Please mention in a footnote the relevant ICD-10 codes for the different groups as well as what “Comorbidities” encompasses. Please define “exposed” and “unexposed”.

2) Figure 1/2: In the figure description, please explain the time frame of the different epochs and provide the relevant ICD-10 codes for the different groups. Please ensure to define all abbreviations used (ICD, ATC).

3) Please be sure to revise the supplementary figures according to the comments on the main figures and tables. Please remember that all figures and/or tables should be self-explanatory on a stand-alone basis.

REFERENCES

1) Where website addresses are cited, when specifying the date of access, please use the word “accessed” instead of “cited” (e.g. [accessed: 10/04/2024]).

2) Please ensure that journal name abbreviations match those found in the National Center for Biotechnology Information (NCBI) databases (http://www.ncbi.nlm.nih.gov/nlmcatalog/journals), and are appropriately formatted and capitalised. For example, “The Lancet Healthy Longevity” in reference [13] should be “Lancet Healthy Longev”.

SOCIAL MEDIA

To help us extend the reach of your research, please provide any X (formerly known as Twitter) handle(s) that would be appropriate to tag, including your own, your co-authors’, your institution, funder, or lab. Please enter in the submission form any handles you wish to be included when we post about this paper.

Comments from Reviewers:

Reviewer #1: I appreciate the efforts made by the authors to address my and my fellow reviewers' constructive comments. Changes have improved the manuscript and increased its impact.

Reviewer #2: We thank the authors for largely addressing our previous comments. The explanation on other medications/treatments might be indicated in the text, possibly as a limitation.

[LINK]

General Editorial Requests

---

## [Editor Report · Decision Letter 3]

31 May 2024

Dear Dr Formánek, 

On behalf of my colleagues and the Academic Editor, Alexander C Tsai, I am pleased to inform you that we have agreed to publish your manuscript "Deaths with COVID-19 and from All-causes following First-ever SARS‑CoV‑2 Infection in Individuals with Pre-existing Mental Disorders: A National Cohort Study from Czechia" (PMEDICINE-D-24-00730R3) in PLOS Medicine.

I appreciate your thorough responses to the reviewers' and editors' comments throughout the editorial process. We look forward to publishing your manuscript, and editorially there are only a few remaining minor points that should be addressed prior to publication. We will carefully check whether the changes have been made. If you have any questions or concerns regarding these final requests, please feel free to contact me at atosun@plos.org.

Please see below the minor points that we request you respond to:

1) l.128: Please add “to our knowledge” or similar (“To our knowledge, no study has used national data…”).

2) ll.435-436: “Higher E-values increase the confidence that the detected associations are not due to unaccounted for confounding.” – could you specify "higher" (e.g. ">1") so that the reader can judge the magnitude of the numbers presented (we suggest adding this in the Methods as well, if possible)?

PRESS

Sincerely, 

Alexandra Tosun, PhD 

Associate Editor 

PLOS Medicine